

# Role of surface hydrology in determining the seasonal cycle of Indian summer monsoon in a general circulation model

Shubhi Agrawal[1] and Arindam Chakraborty[1,2]

[1]Centre for Atmospheric and Oceanic Sciences, Indian Institute of Science, Bengaluru, India
[2]Divecha Centre for Climate Change, Indian Institute of Science, Bengaluru, India

*Correspondence to:* Arindam Chakraborty (arch@caos.iisc.ernet.in)

**Abstract.** Rainfall during summer monsoon season (June–September; JJAS), that accounts for about 80% of the yearly total over Indian region, is herald by its onset over Kerala in June. And these four summer monsoon months contribute 19%, 32%, 29% and 20% of the seasonal rainfall, respectively. Therefore, it is important that this seasonal cycle is captured by general circulation models (GCMs) used to understand and predict monsoon. In this study, using decade-long simulations of

an atmospheric GCM, we show that surface hydrology over India as well as over its surrounding regions plays a central role during the onset phase of monsoon and thus modulates seasonal cycle. The model, in its default configuration, simulates early onset and excess precipitation (about double of that observed) over the Gangetic plain (GP) in June. Moreover, the model has large positive surface soil moisture bias over India throughout the year and negative bias over the arid-semiarid regions to the north-west of India during the pre-monsoon months. From multiple sensitivity experiments, it is discerned that the remote dry

soil moisture bias in the model over the Western Central Asia region intensifies the tropospheric low-level circulation causing excessive moisture advection, followed by moisture convergence over the GP in the beginning of June, and an early onset. Local soil moisture over GP makes a diminutive contribution to precipitation bias in June. But as the season progresses and the remote influence weakens, the increased local soil moisture regulates surface and near-surface conditions which subsequently reduces moisture convergence over GP, reducing precipitation in the later phase of monsoon. The results presented here can be

useful for diagnosis and improvement of land surface models.

## 1  Introduction

India lies on the edge of the bigger south east Asian monsoon regime (Wang and LinHo, 2002) and receives almost 80% of its annual rainfall during June to September. Uncertainties involved in predicting the onset (Joseph et al., 1994) and Indian summer monsoon rainfall (ISMR) is high (Sperber and Palmer, 1996). Slowly varying SST related external factors together

contribute only half of the interannual variability of Indian summer monsoon, limiting its predictability (Ajaya Mohan and Goswami, 2003). Many studies in the past have also emphasised the role of land-atmosphere coupling over Indian region. Interactive soil moisture (SM) improves the model simulation of northward propagation of clouds and active-break cycles, key features of Indian summer monsoon (Rajendran et al., 2002) and the hydrological surface feedbacks affect the temporal characteristics of the Tropical Convergence Zone (TCZ) rather than spatial characteristics (Ferranti et al., 1999). Monsoon





core over Indian region is identified as a 'Hotspot' for coupling by Koster et al. (2004). The contribution of land-atmosphere interaction in regulating the annual cycle of Indian monsoon (Bollasina and Ming, 2012), especially during the onset phase, and the role of surface heating over north-west part of India and west central Asia in creating the longitudinal heating gradient, which plays important role in monsoon progression over India (Bollasina and Nigam, 2011), has been shown. Recent papers,

like Rai et al. (2015) also explores the link between pre-onset rainfall over western central Asia and ISMR and suggests that it is negatively correlated to ISMR, mainly in June-July months. And Samson et al. (2016) show that land-surface-temperature anomaly over the west central Asia region is positively related with the intensity of low-level Westerly jet, which in turn is positively related to ISMR.

Much work has been done on similar lines to understand the more basic soil moisture - precipitation feedback mechanism

underlying the land-atmosphere coupling. Surface albedo and Bowen ratio are functions of soil moisture and soil type and determine the radiation budget at the surface and rationing of solar flux into latent and sensible heat (Eltahir, 1998). The soil moisture value also limits the evaporation from the ground and transpiration by plants. The heat and moisture fluxes from ground into the boundary layer play a significant role in determining the stability conditions of the atmosphere and thus have the potential to influence the atmospheric dynamics (Neelin and Held, 1987). Webster (1983) talks about the role

of sensible heating in destabilising the atmosphere and also about increased stability of atmosphere due to surface cooling through evaporation of soil moisture. Shukla and Mintz (1982), using an atmospheric model, illustrate that the spatial structure of temperature, winds, and precipitation greatly depend on the evapotranspiration from surface. Studies by Zheng and Eltahir (1998), Douville et al. (2001) and Douville (2002) over African region; Betts et al. (1996), Findell and Eltahir (1997) and Eltahir (1998) over North American region established the positive feedback between precipitation and local soil moisture.

Koster et al. (2014) demonstrate, using an atmospheric model, remote effects of surface soil moisture anomalies of a region on atmospheric conditions over another region through planetary wave activity, and also found supporting evidence in reanalysis data.

Similar studies over India offer distinct but sometimes contrasting perspective on the nature of the coupling between precipitation and soil moisture. Meehl (1994) studied the nature of land-atmosphere coupling over Asian region during summer

monsoon season in terms of land albedo effect and soil moisture contributions, and concluded that soil moisture and precipitation are positively coupled, due to increased evaporation from wetter land surface and also that the land-sea temperature gradient is mainly driven by land albedo, lower albedo increases the net radiation over land thus causing warmer land surface. Futami et al. (2009) showed through their sensitivity experiments that the land surface conditions play a critical role in intensifying convection over the Indian region, at the beginning of monsoon, but the coupling effect is not explicit for the

later phase of monsoon. Whereas Saha et al. (2012) in their work showed that land-atmosphere feedback modulated the intraseasonal variability of ISM much after the onset phase as well. Asharaf et al. (2012), using a regional model, showed that soil moisture-precipitation are negatively coupled over the north-west India and positive over the eastern part of India, but they also noted that lower pre-onset soil moisture conditions caused delayed onset over India. Thus, it could be said that the nature of land-atmosphere coupling may vary spatially over Indian region due to its diverse climate regimes and also large spatial



inhomogeneity in terms of soil type, orography, vegetation. Hence there remains a scope to further explore this link in the context of Indian region.

India has vast alluvial plains in the northern and central India formed by rivers like Ganges, Narmada, Mahanadi and their tributaries, which act as a bed for the monsoon trough and the arid and semi-arid regions to the north-west of India where the

surface heating is high; along with the mountains of Himalayas and Western Ghats. These regional features together have a strong impact on the manifestation of Indian summer monsoon. Therefore studying and understanding the land-atmosphere coupling over Indian region is important. In this study we use an AGCM, coupled with a land model, to investigate the role of surface hydrology on the seasonal cycle of the ISM. The paper is organised as following. Section 2 explains the model configuration and observational and reanalysis data used for analysis. It also covers detailed experimental setup details. Sec-

tion 3 covers the critical analysis of the model with respect to observations and reanalysis data, followed by model experiments through which we illustrate the nature of soil moisture and precipitation coupling over Indian and the surrounding region. Overall inter-comparison of all experiments is also presented. Main findings of the paper and future course are summarised in Sect. 4.

## 2   Model and Experiment details

### 2.1   Model description

The Community Earth System Model (CESM) has been developed at National Center for Atmospheric Research in Boulder Colorado. In this study, CESM 1.0.4 with coupled Community Atmospheric Model (CAM 5.1.1) and Community Land Model (CLM 4.0) is used along with data Ocean Model. The CAM model is used with Finite-Volume dynamical core with a grid resolution of 0.43°(latitude) x 0.63°(longitude). The horizontal discretization is based on semi- Lagrangian scheme by Lin

and Rood (1996, 1997) and the vertical discretization is quasi-Lagrangian. It has 26 hybrid levels in vertical direction. The model physics parameterization package used is CAM4 and consists of components to handle moist processes, radiation and clouds, and turbulent mixing. Deep convection scheme is by Zhang and McFarlane (1995) and includes enhanced moist physical representations ensuing Raymond and Blyth (1986, 1992). Moist turbulence scheme for parameterization of boundary layer processes is by Holtslag and Boville (1993) and shallow convection scheme by Hack (1994). Radiative transfer parameteriza-

tion used is CAM Radiative Transfer scheme. Cloud microphysics is by Rasch and Kristjánsson (1998) and calculations for evaporation of convective precipitation is following Sundqvist (1988). Fluxes of heat, momentum, and moisture over the ocean are as per Bryan et al. (1996).

CLM4 (Bonan et al., 2002) has nested subgrid level hierarchy to represent spatial land surface heterogeneity, in which each grid cell can have multiple landunits, soil/snow columns and plant functional types. Landunit types capture broad spatial

patterns like wetland, lake, glacier, urban and vegetated. The columns capture the variability in state variables for energy and water and their fluxes within a single landunit. The column has 15 layers in vertical to represent soil and 5 layers of snow. The soil type is determined by its colour (Lawrence and Chase, 2007) and the ratio of sand, clay and organic matter. Radiative transfer within vegetative canopies is described in Bonan (1996). The ground albedos are a combination of soil and snow





albedos where soil can belong to any landunit type. Heat and momentum fluxes from ground are estimated using Monin-Obukhov Similarity theory. Soil hydrology is handled by an extensive and sophisticated set of modules and is implemented for top 10 soil layers. Bottom 5 layers are considered hard rocks. Processes like interception, through fall, canopy drip, infiltration, surface runoff, evaporation, exchanges within soil column, etc are parameterized to calculate the changes in soil moisture (Zeng

and Decker, 2009), canopy water, snow water and aquifer. The lake model is by Zeng et al. (2002).

The CAM model, coupled with CLM model, is run with monthly climatological SSTs. The sea-ice component has been run in prescribed mode where ice coverage is prescribed. Land-ice component is in stub mode.

## 2.2   Observational Datasets

Satellite-based merged-IR precipitation product, TRMM-3B42, version-7 (Huffman et al., 2007), is used. The three-hourly

dataset is available at 0.25°x 0.25°spatial resolution. Soil moisture dataset titled 'ESA CCI ECV Surface Soil Moisture Combined Product' is used and referred as ESACCI hereafter. This dataset is a combined product of active and passive sensors (Wagner et al., 2012; Liu et al., 2012) and is available since 1978 until the recent year 2013 at daily temporal resolution. Its spatial coverage is from -180°to 180°E and -90°to 90°N, at a resolution of 0.25°x 0.25°. The data represents a soil depth of 0.5-2 cm. Validation of this dataset has been carried out by Dorigo et al. (2015). Apart from this, daily data from ERA-Interim

(Dee et al., 2011) at a spatial resolution of 0.75°x 0.75°is used for specific humidity, horizontal winds, surface pressure, surface evaporation fields. Henceforth it is denoted as ERA. The data for period 2000–2010 is used for this study for all the observational and reanalysis products because of good spatial and temporal resolution.

## 2.3   Experiment Setup details

Experiments conducted with the model configuration described above are explained here and are briefed in Table 1.

**Control Run:** CAM5 model, with the configuration as explained in Sect. (2.1), is run for 10 years with default model code, forced with climatological SSTs from HadISST dataset (Rayner et al., 2003) for the period 1982–2002. This experiment is referred as *CNTL* henceforth.

**Globally Modified Runs:** Two runs with the same model configuration as in *CNTL*, but modified soil hydrology (in CLM), are made. In the first run, referred as Global Dry Run (GLDRY), volumetric soil moisture in all 10 soil layers over land is

fixed at 1% of soil water saturation value. Second run, Global Wet Run (GLWET), is performed on the similar line except that volumetric soil moisture is fixed at 100% soil water saturation value. Excess or deficit water is adjusted with water from aquifer to close water budget.

**Regionally Modified Runs:** Three runs with the same model configuration as in *CNTL* and similar modifications in soil hydrology as in globally modified runs, but over smaller regions, are made. In the first run, referred as Gangetic Plain Dry

Run (GPDRY), volumetric soil moisture over Gangetic Plain region (74°–88°E, 20°–28°N) is kept fixed at 1% of saturation values for all the 10 layers. The second run, referred as Gangetic Plain Wet Run (GPWET), is similar to GPDRY run, except that volumetric soil moisture is kept fixed at 100% saturation values. In the third run, referred as West Central Asia Wet Run (WCAWET), volumetric soil moisture over west central Asian region (40°–72°E, 23°–45°N) is kept fixed at 100% saturation





values for all the 10 layers. Over the rest of the domain, soil moisture is allowed to vary in response to the atmospheric and land processes.

**Regionally Nudged Runs:** Three more runs with the same model configuration as in *CNTL* are made, with changes in soil hydrology. This time instead of fixing the soil moisture to some prescribed value, soil moisture is nudged towards observational values at each time-step over the chosen domain with a relaxation parameter comparable to time-step used to run model, that is thirty minutes. The observation values are obtained from ESACCI soil moisture dataset described in Sect. (2.2) and climatological monthly means are calculated over the chosen region. Soil moisture of top two model layers is nudged, as the observational dataset represents surface soil moisture. In the first run, referred as Gangetic Plain Nudged Run (GPNDG), soil moisture over the extended Gangetic Plain region (72°–89°E, 22°–31°N) is nudged using observations. In the second run, referred as West Central Asia Nudged Run (WCANDG), soil moisture over west central Asia (40°–72°E, 23°–45°N) is nudged. In the third run, referred as West Central Asia-Gangetic Plain Nudged Run (WCAGP_NDG), soil moisture over both above mentioned regions is nudged using the observed soil moisture values from respective region. All runs are 10 years each. Daily output is saved and in the present study 10-year ensemble means are presented.

## 3 Results

### 3.1 Rainfall climatology of *CNTL*

Climatologies of rainfall during boreal summers (June-September: JJAS) over the tropics from satellite-based observations (TRMM) and the CAM control simulation (*CNTL*) are shown in Fig. 1a and b. We recognise that the major centres of Asia-Pacific summer monsoon (Wang and LinHo, 2002) — the South Asian, East Asian, West Pacific, East Pacific, and Indonesian monsoon domains — are captured reasonably well by *CNTL*. However, notable differences being, *CNTL* simulates weaker than observed rain intensity over tropical and subtropical western Pacific Ocean and stronger than observed rainfall over equatorial central Pacific Ocean. The pattern correlation coefficient (cc) between TRMM and *CNTL* for the region (15°S– 45°N, 40°–290°E) shown in Fig. 1a is 0.76.

A careful look at Fig. 1 shows that the prominent features of summer monsoon, like heavy rainfall along the Western Ghats, the foothills of the Himalayas, and over the monsoon trough are captured correctly by *CNTL*. However, the rain intensity and the spatial extents of these rainy regions are overestimated. On the contrary, intense rainfall zones along the Myanmar coast of Bay of Bengal are underestimated. We emphasise here that, the regions with strong rainfall bias in the model, $viz.$, the western coast of peninsular India, northern Bay of Bengal and Himalayan foothills, are influenced by orography. Convection over these regions is highly sensitive to speed and direction of low-level horizontal winds.

Taken as a whole, the observed seasonal mean features are present in the model *CNTL* simulation. Similar spatial features of simulated rainfall over Indian region have been reported earlier by Meehl et al. (2012) and Islam et al. (2013) for atmosphere-only and ocean-coupled versions of the CESM.

In Fig. 1c, climatological annual cycles of rainfall from TRMM and *CNTL* averaged over Indian region (8°–28°N, 70°–90°E, land part) are compared. We find that the model grossly overestimates rainfall in June ($\sim$77%) and remains moderately





higher (∼28%) in July–September (JAS). In addition, *CNTL* also overestimates rainfall of post-summer monsoon (October-November) as well as winter monsoon seasons (December–February). However, the rainfall intensities are comparable during the pre-monsoon months (March–May). Interestingly, the large bias in *CNTL* in June is reduced substantially during the later months (JAS) of the summer monsoon.

How is the positive bias in *CNTL* rainfall over Indian region spatially distributed and how do the differences compare between early (June) and late (JAS) phases of the monsoon? This question leads us to plot the spatial pattern of difference between *CNTL* and TRMM for June and JAS separately. As pointed out previously, the high bias along the orographic regions of Himalayas, Western Ghats, and Myanmar coast persists in both June and JAS, as can be perceived from Fig. 2a, b. But there is a huge contrast in the rainfall for June and JAS over northern Gangetic plains of India, as pointed out from the overlaid box

(76°–88°E, 22°–28°N), henceforth referred as Gangetic Plains (GP). While *CNTL* heavily overestimates rainfall over GP in June, contrarily it underestimates the rainfall over GP in JAS, specially towards the eastern side of GP. This switch in the nature of bias over GP from June to JAS leads us to further investigate the precipitation climatology over the region and the factors influencing it.

Figure 2c shows climatological monthly mean rainfall, area averaged over GP, for *CNTL* and TRMM. While in June *CNTL*

overestimates rainfall by about 85%, JAS mean rainfalls are comparable. This change includes a dramatic decrease in *CNTL* rainfall bias over GP from June to July. Since June rainfall has a high correlation with the onset date, we next investigate the daily climatology of rainfall over GP (Fig. 2d). High rainfall in June over GP in *CNTL* can be attributed to the sharp rise in daily rainrate in the beginning of June, which stays high for the remaining month. However, in observations daily rainrate increases in mid-June and attains peak in mid-July. We calculate onset date of monsoon over GP for *CNTL* and TRMM following the

criterion used by Chakraborty et al. (2006), that is an onset is declared if the area averaged rainfall is more that 4 mm day$^{-1}$ for at least five consecutive days after first of June. The mean onset date over GP in *CNTL* is fifth June with an interannual standard deviation of 5 days, whereas in TRMM, it is 13$^{th}$ June with an interannual standard deviation of 10 days. This lower interannual variation in onset date in *CNTL* compared to TRMM could be due to use of climatological SST in *CNTL* run. However, here we emphasise that the onset date in *CNTL* is substantially advanced which results in a peak of seasonal cycle in

June, one month early compared to TRMM.

In order to qualify this excess (comparable) rainfall in *CNTL* as against TRMM estimates during June (JAS), we examine and compare the atmospheric moisture budget from *CNTL* and ERA. During summer monsoon, the low-level winds over the Arabian Sea and the Bay of Bengal bring moisture into the Indian region (Joseph and Raman, 1966; Findlater, 1969). Vertically integrated (surface to 100 hPa) moisture flux vectors (in kg m$^{-1}$ s$^{-1}$), calculated for June and JAS, from *CNTL* and ERA are

shown in Fig. 3. Superimposed on it is moisture divergence (kg m$^{-2}$ s$^{-1}$) in shades. We find a stronger incoming moisture flux toward Indian region during June in *CNTL* than ERA. For clarity, moisture fluxes through the four walls of GP box, in units of mm day$^{-1}$, are shown with bold arrows. Higher zonal moisture convergence in June is noticeable in *CNTL*. In JAS (Fig. 3 c and d), in *CNTL*, most of the moisture flux that enters GP from eastern and southern boundaries crosses over to the Himalayan foothills through northern boundary, without converging. Whereas, in ERA higher zonal moisture convergence over

GP can be noted. Area-averaged values of precipitation (P) and evaporation (E) (in mm day$^{-1}$) over GP are mentioned in upper



right corner of the figure. We find that June mean rainfall in *CNTL*, which is nearly double compared to observation, can be attributed to almost double net moisture advection (convergence) (6.8 mm day$^{-1}$) in *CNTL* compared to ERA (2.9 mm day$^{-1}$). Evaporation is only slightly higher in *CNTL*. In JAS, the moisture convergence over GP reduces to 4.6 mm day$^{-1}$ in *CNTL*, whereas for ERA it increases to 3.8 mm day$^{-1}$ from June, showing the weakening (strengthening) of moisture convergence

in JAS in *CNTL* (ERA). Evaporation in JAS is still slightly higher at 3.7 mm day$^{-1}$ in *CNTL* compared to 2.8 mm day$^{-1}$ in ERA, hinting at bias in *CNTL* in local soil moisture conditions. Note that, IMD precipitation data (Rajeevan et al., 2006) has been used along with ERA moisture flux and evaporation data for moisture budget calculations. Thus we do not expect perfect closure of moisture budget. However, in *CNTL*, the moisture budget closes with an error margin of -0.5 to 0.5 mm day$^{-1}$. Also noteworthy is the difference between *CNTL* and ERA in terms of moisture divergence (shaded positive-values)

over the north-west region, and the leeward side of Western Ghats and the Himalayas. *CNTL* does not capture this divergence and overestimates rainfall in these regions.

We know that the convergence of moisture over a region is largely influenced by the atmospheric and near-surface conditions of the region (Neelin and Held, 1987; Srinivasan and Smith, 1996). Vertical moist stability (VMS, Eq. 2), that is the difference of moist static energy (MSE) between top layer (500 hPa to 100 hPa, Eq. 3) and bottom layer (surface to 500 hPa, Eq. 4), is a

good measure of energy available for convection and atmospheric instability (Chakraborty et al., 2014). Figure 4 presents, for June and JAS, variation of VMS along the longitude 60°–90°E over northern belt of India (22°–28°N) and moist static energy (MSE, Eq. 1), moisture ($L\,q$) and temperature ($C_p\,T$) terms for lower atmospheric layer (surface to 500 hPa level) for *CNTL* and ERA. VMS gradient along longitude for ERA and *CNTL* is shown in Fig. 4a and e respectively, for June (solid curve) and JAS (dashed curve). It can be noted that the distribution of VMS along longitude is similar for June and JAS in ERA, that is it

increases from east to west over GP, signifying more instability towards eastern side and an increase in atmospheric stability as we move westwards. But in *CNTL* June and JAS mean VMS distribution is different. In June it increases from east to west, similar to ERA, but in JAS the gradient in VMS along longitude is very small and in fact VMS is slightly lower on the western side compared to the eastern side. This shows that the atmospheric stability conditions change considerably in *CNTL* as the season progresses from June to JAS, with eastern side gaining stability and western side becoming more unstable.

The transition in VMS structure from June to JAS in *CNTL* can be explained through changes in lower level MSE. In ERA, lower layer MSE (Fig 4b) decreases from east to west for both June and JAS, with JAS MSE slightly higher than June. It can be mainly attributed to moisture term (Fig. 4c), with more (less) moist lower layer towards east (west), which parallels with lower (higher) air temperature towards east (west) (Fig. 4d). But in *CNTL* lower layer MSE (Fig. 4f) is higher in June compared to JAS towards the eastern side, resulting in lower VMS in June towards eastern side. In JAS, however, the lower layer MSE

becomes almost same across longitude and the gradient of MSE is negligible, explaining the similar pattern of VMS. On the western side, the increase in moisture term (Fig. 4g) supersedes the drop in temperature term (Fig. 4h), resulting in net increase of MSE in the lower layer by approximately 2 kJ kg$^{-1}$. Whereas, on the eastern side, though there is an increase in moisture term, but the reduction in temperature term is higher, resulting in a net decrease of MSE. It can be inferred that the increase in moisture near-surface causes a drop in near-surface temperature in JAS, ultimately increasing the stability of atmosphere over

GP.



Lower layer MSE is directly influenced by the surface conditions and exchange of fluxes of heat and moisture between ground and the boundary layer. Surface soil moisture has been identified as an important factor that can influence the surface temperature distributions (Berg et al., 2014) and also Bowen ratio that determines the rationing of net solar radiation into sensible ($C_p\ T$) and latent ($L\ q$) heat fluxes (Eltahir, 1998; Delworth and Manabe, 1989). Hence we compare the model

simulated surface soil moisture (SM) with recently available satellite-based ESACCI observations. Figure 5 a-c shows spatial distribution of volumetric surface SM for June mean climatology of (a) ESACCI, (b) *CNTL*, (c) Spatial difference between them (*CNTL* − ESACCI). It can be observed that though the spatial pattern of surface SM is simulated well by model, but the Indian region has high positive SM bias, while the surrounding regions, specially arid-semiarid regions to the west of India have negative SM bias for June. We further present the monthly climatology of SM over GP and West Central Asia region

(WCA; 40°–72°E and 23°– 45°N) from *CNTL* and ESACCI in Fig. 5d and e. The annual cycle of SM over GP is captured well by *CNTL*, but the monthly means are overestimated throughout the year and the bias peaks in the month of June and remains substantially high for the subsequent months (JAS). Whereas for WCA, *CNTL* does not capture the observed annual cycle well and SM bias in *CNTL* changes its sign from negative during the pre-monsoon months (April-July) to positive August onwards.

This soil moisture bias pattern, that is, positive bias over GP and negative bias over WCA in *CNTL* can generate surface

temperature and pressure anomalies, which might have extensive impact on the onset phase of monsoon over India and its seasonal cycle. In fact, in a recent work, Samson et al. (2016) show that surface temperature anomalies are a dominant source of bias in the simulated low-level circulation and precipitation over Indian region in GCM. Lower surface temperature over the Middle-East region weakens the low-level westerly jet, while higher surface temperature over India intensifies the monsoon circulation. We have also shown in the above analysis that substantial changes occur in atmospheric stability through changes

in lower layer MSE when *CNTL* slightly overestimates surface evaporation over GP. Through regionally modified runs, we show how the local and remote soil moisture biases over GP and WCA influence the monsoon circulation and rainfall over GP. Further, we attempt to explain the tender balance between local evaporation and moisture convergence that contributes to the total precipitation bias in June over GP in *CNTL*.

### 3.2 The local forcing

The influence of local soil moisture conditions on the precipitation simulation over GP is analysed through three experiments, with regionally modified SM over the Gangetic Plain region. These are GPDRY, GPWET, and GPNDG and their model setup details are explained in Sect. (2.3). Through GPDRY and GPWET runs, we analyse the effect of local SM on the precipitation simulation over GP under extreme conditions, that is very dry surface and completely saturated surface. Through GPNDG, presented later in the section, we analyse the local coupling under interactive but nudged SM conditions. Figure 6 shows the

spatial pattern of difference between simulated precipitation and 850 hPa winds from these regionally modified runs and *CNTL* in June and JAS, with precipitation difference significant at 95% level dotted. From Fig. 6a, it can be observed that under dryer local SM conditions, low-level wind circulation is more cyclonic over GP in GPDRY in June compared to *CNTL*, causing higher moisture advection. But in spite of this rainfall shows small negative anomaly over GP which is counter-intuitive. In JAS (Fig. 6d), both cyclonic circulation and rainfall increases over GP in GPDRY compared to *CNTL*. On the contrary, for





locally saturated SM conditions in GPWET, the cyclonic circulation weakens (Fig. 6b) and also rainfall decreases in June. But there is no significant change in JAS (Fig. 6e).

Moisture budget calculations over GP presented in Fig. 7 further clarifies the local impact. In June (Fig. 7a), for GPDRY, net moisture advection (Net_Adv) slightly increases and evaporation (E) reduces substantially, overall reducing precipitation

(P) compared to *CNTL*. On the contrary, for GPWET, net moisture advection reduces substantially and evaporation increases slightly , again reducing the precipitation compared to *CNTL*. It can be inferred from here that, there is a trade-off between local evaporation and net moisture advection from surrounding over GP. Recall from the comparison between observations and *CNTL* in Fig. 3a and b, that in *CNTL* both evaporation and net moisture advection over GP (3.5 mm day$^{-1}$, 6.8 mm day$^{-1}$ respectively) in June are higher compared to ERA (2.5 mm.day$^{-1}$, 2.9 mm.day$^{-1}$). Thus, though the small positive

bias in evaporation can be attributed to local SM bias in June in *CNTL*, but the excess moisture advection over GP can not be explained in presence of such local SM bias. However, such persistent SM bias can influence the circulation in later part of season. In JAS (Fig. 7b), it can be seen, for drier SM conditions (GPDRY) net moisture advection and net zonal moisture advection (Zn_in) increase considerably, whereas for wetter SM conditions (GPWET) they are comparable to *CNTL*. This asserts that excess SM over GP in JAS in *CNTL* weakens the circulation and the surface conditions are closer to saturation.

The effect of local surface conditions on atmospheric stability is further elaborated. Vertical profiles of MSE, moisture term ($L\,q$), temperature term ($C_p\,T$) are calculated for GP for June and the differences with respect to *CNTL* is presented in Fig. 8 for a) GPDRY, b) GPWET, and c) GPNDG. Note that the basic difference between these experiments and *CNTL* is only soil moisture conditions. Yet we notice considerable differences in MSE profile at ∼750 hPa with respect to *CNTL*, which represents the low-level moisture convergence (see moisture curve). Clearly, the low-level moisture convergence increases (decreases) in

GPDRY (GPWET), increasing (decreasing) the lower MSE and consequently decreasing (increasing) VMS.

GPDRY and GPWET experiments have fixed SM conditions, hence one more experiment (GPNDG) with interactive SM conditions is demonstrated to ascertain our results. From Fig. 6c and f , it can be noted that nudging local SM has a significant impact on not just GP but also on northern region of the Arabian Sea and the Bay of Bengal. Over GP rainfall decreases in June and increases in JAS. The decrease over central Arabian sea and central India in JAS is significant. From the moisture

budget analysis (Fig. 7), it becomes explicit that the decrease in June precipitation is due to decrease in evaporation and net moisture advection remains unaffected, hinting at local SM bias contribution. In JAS, circulation and moisture convergence over GP increase, increasing rainfall slightly compared to *CNTL*. Also from Fig. 8c, it can be noticed that the change in MSE in June is mostly close to surface level owing to local soil moisture nudging. Overall the low-level MSE decreases for GPNDG, increasing the VMS slightly in June compared to *CNTL*.

To summarise, local positive soil moisture bias contributes to precipitation bias in June, through evaporation, but can not explain the intensification of circulation and excess moisture convergence over GP in *CNTL*. This confirms the existence of a factor that remotely influences the manifestation of Indian monsoon during the onset phase. In JAS, however, local SM bias can be held responsible for the weakening of monsoon circulation over GP in *CNTL* and reduction in precipitation through changes in atmospheric stability and lower layer MSE. In the next section, we analyse the influence of negative soil moisture





bias over the arid-semiarid regions of Afghanistan, Pakistan, and Iran, that lie to the north-west of India and understand the nature of the coupling between rainfall over GP and SM anomalies over western central Asia (WCA).

## 3.3 The remote forcing

Previously in Fig. 5c and e, it has been shown that Western Central Asian region (WCA) has negative soil moisture bias
compared to observations during the pre-monsoon months. This can have serious implications during the onset phase (June) of the Indian monsoon and thus we investigate the same here. Two experiments are conducted to study the remote effect. As *CNTL* has negative SM bias over WCA, one experiment with wetter conditions over WCA is conducted, WCAWET, where SM is kept fixed at saturation values for WCA. Another experiment (WCANDG) is conducted where surface SM over WCA is nudged with observed monthly mean SM values from ESACCI (Experimental setup details are presented in Sect. 2.3).

Figure 9 shows the spatial pattern of difference between simulated precipitation and 850 hPa winds from regionally modified runs and *CNTL* for June and JAS, with rainfall difference significant at 95% level dotted. It can be noted that June rainfall and cyclonic wind circulation over GP reduce substantially in WCAWET (Fig. 9a), indicating strong remote influence of surface conditions over WCA on rainfall over GP. Also, rainfall decreases substantially over the north Arabian sea and increases over southern part of Arabian sea and Bay of Bengal, suggesting an equatorward shift of low-level westerly Jet in WCAWET
compared to *CNTL*. This emphasises that the intensity and northward extent of low-level jet in June are sensitive to the SM conditions over WCA. On the other hand, in JAS (Fig. 9c), WCAWET and *CNTL* are almost comparable, hinting at the possible weakening of the remote influence over GP in later part of the season. Again, we ascertain our findings through nudged experiment, WCANDG. Similar results are obtained, that is, weakening of circulation over GP in June and thus a reduction in rainfall. Not much difference is seen in JAS compared to *CNTL*.

The remote influence on the moisture convergence over GP can be asserted from moisture budget analysis over GP. In June (Fig. 10a), all three moisture components — precipitation, net advection and evaporation— reduce considerably in WCAWET compared to *CNTL*, with a prominent reduction in net advection term. WCANDG shows smaller changes in net moisture advection and evaporation term, and thus a small decrease in precipitation, which is justifiable as the SM nudging is done over small remote area. However, for both WCAWET and WCANDG little changes are observed for JAS, strongly suggesting a
weakening of remote influence on GP as the monsoon season establishes over India.

   The reason for this reduction in rainfall in June in WCAWET becomes more explicit from Fig. 11a, where difference in vertical profiles of MSE, moisture, and temperature terms with respect to *CNTL* are shown for June over GP. MSE drastically decreases for the lower layer of atmospheric column in WCAWET, which can be attributed to the reduction in the moisture term. It makes the atmosphere much more stable as VMS increases for the atmospheric column. The reduction in moisture in
lower layer of the atmosphere can be explained by the weakening or southward shift of the low-level jet, as already seen in Fig. 9a. For WCANDG (Fig. 11b), the change in vertical profile of MSE is small but shows same nature as WCAWET.

   It can be inferred that remote forcing is much stronger compared to local forcing in June. Negative soil moisture anomalies over WCA in *CNTL* results in intensification of low-level westerly jet, which brings in more moisture over GP in June. This moisture increases MSE of the lower level, which in turn makes atmospheric column more unstable, consequently causing



moisture convergence and excess rainfall. However, as the season progresses, the remote influence weakens, and the monsoon is governed by large-scale dynamics and local surface conditions to some extent.

## 3.4 The combined forcing

We demonstrate one final experiment to show improvement in model simulation after correcting both local and remote SM biases. The areas nudged are very small compared to model's global domain as well as the areas over which the biases are observed in *CNTL*. Nevertheless, the results obtained from the nudging run looks very promising, owing to the importance of land-atmosphere coupling over the regions of Gangetic Plains and Western Central Asia, and are summarised here. The spatial pattern of rainfall and 850 hPa winds differences between WCAGP_NDG and *CNTL* in June and JAS are shown in Fig. 12a and b. In June, a significant reduction in rainfall over GP and a southward shift in low-level jet can be seen in WCAGP_NDG. Also, rainfall increases significantly over south-eastern Arabian sea and Bay of Bengal indicating same— the southward shift in ITCZ along with low-level jet. Whereas in JAS, a small but significant decrease in rainfall is seen over south-eastern Arabian sea, and 850 hPa wind difference indicate a slight strengthening of circulation over northern India. In Fig. 12c monthly mean rainfall over GP is presented. June bias reduces considerably in WCAGP_NDG and July rainfall also increases slightly. Overall the seasonal cycle is simulated better in WCAGP_NDG. In Fig. 12d we present daily rainfall time series. In WCAGP_NDG daily mean rainfall increases much more gradually in June compared to *CNTL* and peaks in mid-July only, like in observation, as seen earlier in Fig. 1d. Further, mean onset date got delayed in WCAGP_NDG to ninth June, with an increased standard deviation of 8 days. The results from WCAGP_NDG are satisfactory and changes with respect to *CNTL* are statistically significant.

Finally, we present the overall picture in Fig. 13. As shown earlier, WCA–GP soil moisture anomalies introduce significant circulation changes at low-level and alter the moisture convergence and atmospheric stability over GP. In that light we inspect four important variables for June— (1) Pressure difference index (PDI)- difference of surface pressure (mb) between small region over WCA (55°–72°E and 25°–35°N) and Indian Ocean (55°–72°E and 10°S–0°N), (2) Net moisture advection (Net Flux_IN) into GP, (3) Vertical moist stability (VMS) over GP, (4) precipitation over GP. The difference with respect to *CNTL* is calculated for all the experiments of these four quantities and compared. We observe that GLDRY and GLWET (globally modified runs, described in Sect. 2.3) mark the upper and lower bounds of anomalies in Fig. 13 and all other experiments lie within these extreme bounds. This provides some means of quantification of the sensitivity of precipitation to global soil moisture conditions. From Fig. 13a, which shows PDI versus net advection, it can be noted that PDI is high for GLDRY, low for WCAWET, lowest for GLWET, and thus seems sensitive to SM anomalies over WCA. Net advection varies almost linearly with PDI, with a strong correlation coefficient (cc) of 0.84. Further from Fig. 13b, which shows precipitation versus net advection, we find that though they vary almost linearly, but the correlation coefficient is slightly less, at 0.75. This points to the fact that all the moisture advected into the region need not precipitate out over the region. For example in WCAWET and GLWET, net advection decreases by nearly similar amount, but precipitation is much lower in the case of GLWET. This can be explained from Fig. 13c, which shows that precipitation is strongly correlated with VMS, correlation coefficient being $-0.91$. That is, precipitation is high when VMS is less. VMS is much higher in GLWET compared to WCAWET, which can be attributed to more widely present saturated soil moisture conditions in GLWET, thus precipitation decreases much more.





Another important pair— GPDRY and GPWET— shows the effect of local surface condition on net advection and VMS over GP in June, but the impact is smaller compared to the remote influence (WCAWET). The final experiment WCAGP_NDG falls approximately between GLDRY and GLWET, with a decrease in net moisture advection and an increase in VMS, overall decreasing precipitation over GP.

## 4   Summary and conclusions

In this study, we addressed the influence of local and remote land-atmosphere interactions on the onset phase and seasonal cycle of the Indian summer monsoon, using framework of a general circulation model. In default configuration (*CNTL*), model simulated early onset over Gangetic Plain of India (GP) that resulted in excess (about double of that observed) precipitation in June. This skewed the seasonal cycle of monsoon in *CNTL* with an annual peak in June instead of July as observed. We found that excessive moisture advection and its convergence over GP were the main reasons for this June precipitation bias, where local evaporation contributed minimally. During the ensuing monsoon months (July-September or JAS), the mean precipitation intensity from *CNTL* was comparable to that observed. The excess moisture convergence over GP in June was attributed to decreased vertical stability of the atmosphere caused by increased moist static energy (MSE) of its lower layers. This instability favoured convection, lifted the advected moisture upward in the atmosphere, that in turn increased net convergence. The atmospheric stability over GP was substantially increased during the following months (JAS) due to decrease in lower level MSE that reduced precipitation. As more moisture was added to ground after precipitation, surface temperature decreased due to latent heat loss through evaporation. The decrease in temperature overtook the increase in moisture of lower troposphere and decreased MSE.

Further, we showed that the *CNTL* model simulated higher surface soil moisture compared to observations (ESACCI) over Indian region throughout the year. Whereas, it underestimated the surface soil moisture over surrounding arid and semi-arid regions to the west of India — mainly over west central Asia (WCA) —with higher bias during the pre-monsoon and early monsoon months (March to July). We designed several perturbed model simulations to understand impacts of this local-wet vs remote-dry surface conditions in the model on Indian monsoon.

Through regionally modified model experiments (over GP), we demonstrated the role of local land-atmosphere coupling on seasonal cycle in this model. We showed that the positive local soil moisture bias over GP made a diminutive direct contribution to the precipitation bias in June through evaporation. But local soil moisture influenced the trade-off between evaporation and moisture convergence, through changes in atmospheric stability over the region. Under drier local surface conditions, moisture convergence and atmospheric instability were higher compared to that under wetter local surface conditions. During JAS, precipitation increased substantially in the experiment with drier soil condition as low-level circulation strengthened. However, the June precipitation bias in *CNTL* could not be explained from local surface conditions, as *CNTL* had both high moisture convergence, as well as high evaporation in June.

To understand this mystery further, we designed experiments modulating soil moisture over WCA. We established that the negative soil moisture anomaly over WCA increased pressure gradient between WCA and the equatorial Indian Ocean in June.



This lead to an intensification of the low-level westerly jet, bringing in more moisture to the interior regions of India. The increase in low-level moisture increased MSE and destabilised the atmosphere, resulting in enhanced precipitation in June.

In one final experiment where soil moisture was nudged in the model towards observed values over GP and WCA simultaneously, we noted an improvement in the model simulation of onset and seasonal cycle of precipitation over GP. This was

5    because of a substantial reduction in June precipitation compared to *CNTL*. Hence, it can be concluded that surface conditions over both GP (local) and WCA (remote) modulate the onset phase and seasonal cycle of Indian summer monsoon.

It follows from our work that the surface soil moisture anomalies bear serious consequences on the manifestation of Indian summer monsoon and its onset, and thus a strong need arises to critically analyse and correct soil moisture bias in the land model (CLM), especially over Indian and Asian region. The mechanism proposed here can be invoked to understand the effect

10    of anthropogenic changes induced over Gangetic Plains in recent decades.

*Author contributions.* Both authors conceptualized and designed the study, and interpreted results. Model experiments are carried out by SA.

*Acknowledgements.* AC wish to acknowledge the support received from Ministry of Earth Sciences (MoES)/ Natural Environment Research Council (NERC). SA wish to acknowledge National Center for Atmospheric Research (NCAR) for the use of CESM model.



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



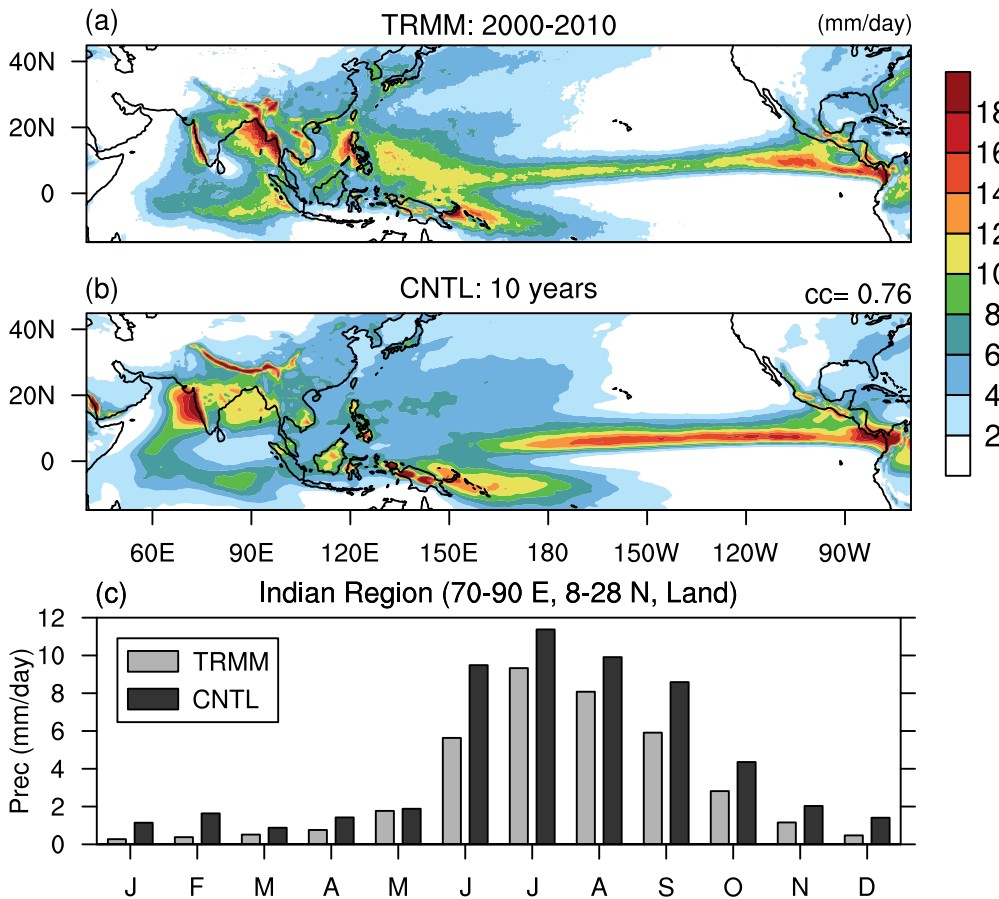

**Figure 1.** Precipitation climatology (in mm day$^{-1}$) from TRMM and *CNTL*. Spatial pattern in JJAS from (a) TRMM and (b) *CNTL*. The pattern correlation coefficient between TRMM and *CNTL* is 0.76. (c) Seasonal cycle over Indian region (70°–90°E, 8°–28°N, land part).




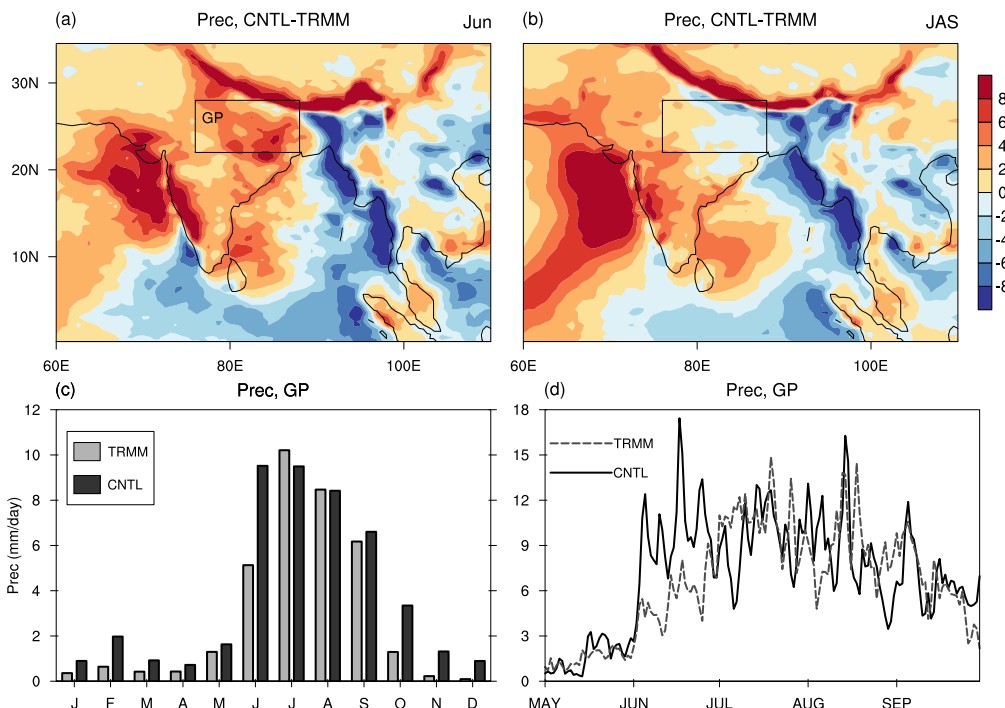

**Figure 2.** Observed (TRMM) and *CNTL* rainfall (in mm day$^{-1}$). Spatial variation of difference between TRMM and *CNTL* rainfall in (a) June and (b) July–September. (c) Monthly mean climatological seasonal cycle and (d) daily time series over Gangetic plain (76°–88°E, 22°–28°N).





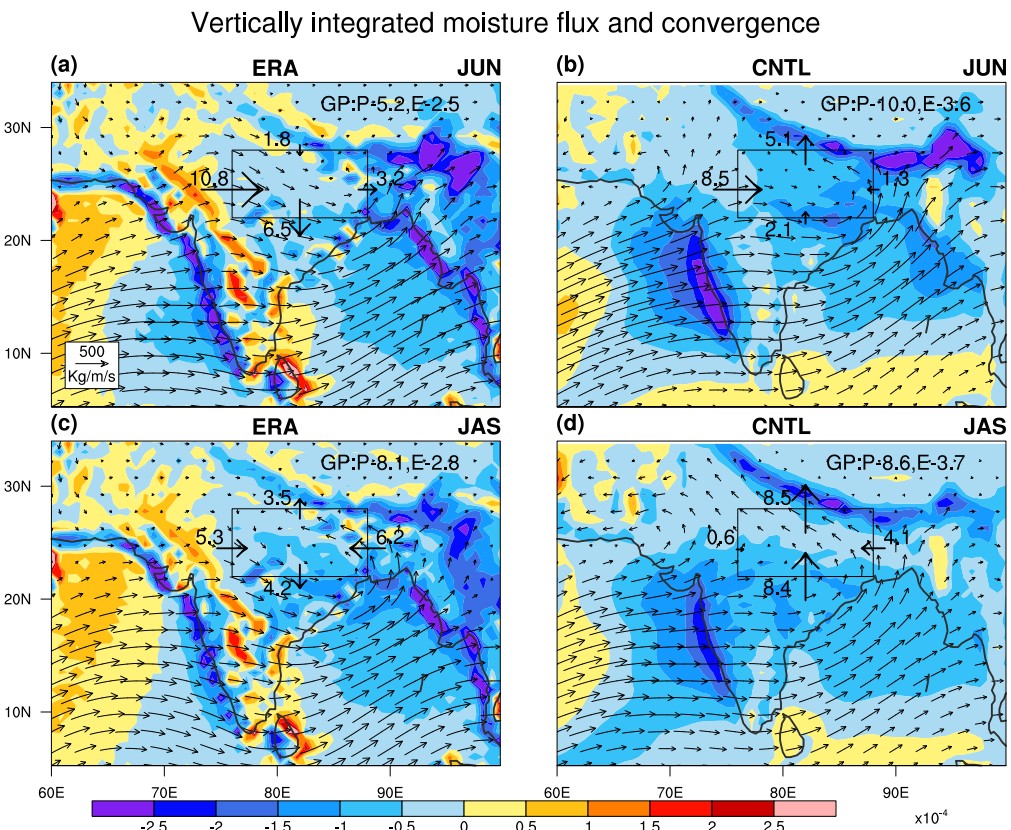

**Figure 3.** Spatial pattern of vertically integrated moisture flux vectors (Kg m$^{-1}$ s$^{-1}$) for June and JAS for ERA and *CNTL*, with moisture convergence (Kg m$^{-2}$ s$^{-1}$) shown in color shades; (a) ERA, June (b) *CNTL*, June (c) ERA, JAS (d) *CNTL*, JAS. Moisture budget components–precipitation (P), evaporation (E) and moisture flux vectors through boundaries of GP (box) are also shown and are in units mm day$^{-1}$.





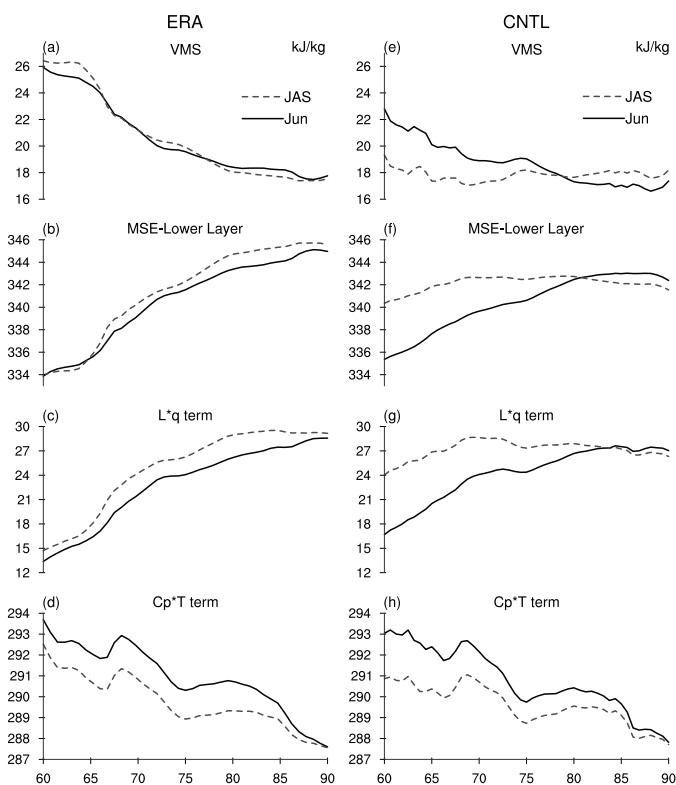

**Figure 4.** VMS of atmospheric column and MSE, moisture ($L\,q$) and temperature ($C_p\,T$) term for lower layer (surface to 500 hPa), averaged latitudinally over $22°–28°$N, is plotted against longitude ($60°–90°$E) for June (solid line) and JAS (dashed line) for ERA and *CNTL*– a) ERA– VMS, b) ERA– MSE for lower layer, c) ERA– $L\,q$ for lower layer, d) ERA– $C_p\,T$ for lower layer, e) *CNTL*– VMS, f) *CNTL*– MSE for lower layer, g) *CNTL*– $L\,q$ for lower layer, h) *CNTL*– $C_p\,T$ for lower layer.



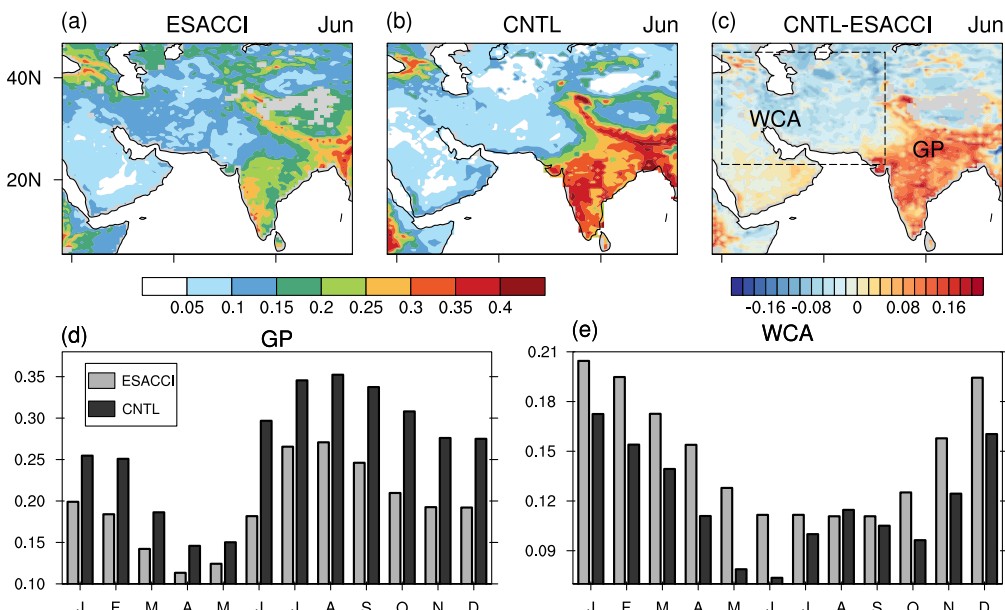

**Figure 5.** June mean spatial pattern of volumetric surface soil moisture from observation (ESACCI) and *CNTL* and their difference: a) ESACCI, (b) *CNTL*, c) *CNTL* −ESACCI. Monthly mean SM from ESACCI (gray) and *CNTL* (black), area averaged over d) Gangetic Plains (GP), e) West Central Asia (WCA).




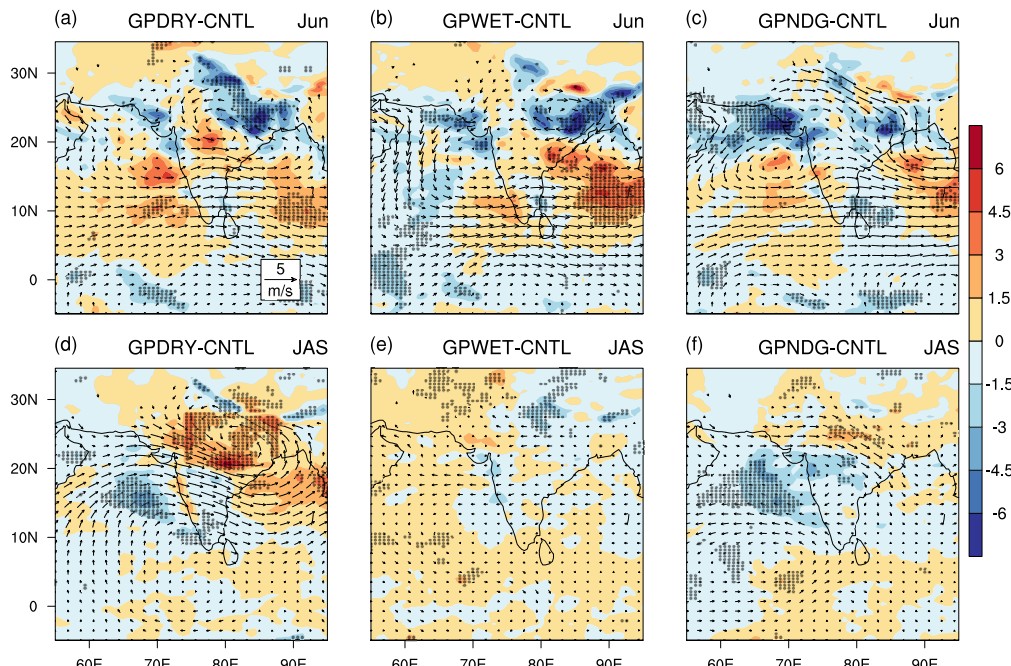

**Figure 6.** Spatial pattern of difference between simulated precipitation (mm day$^{-1}$) and 850 hPa Winds (m s$^{-1}$) from regionally modified (over GP) runs and *CNTL* for June and JAS, a) June, GPDRY−*CNTL* b) June, GPWET−*CNTL* c) June, GPNDG−*CNTL* d) JAS, GPDRY−*CNTL* e) JAS, GPWET−*CNTL* f) JAS, GPNDG−*CNTL*. Precipitation difference significant at 95% level is dotted.




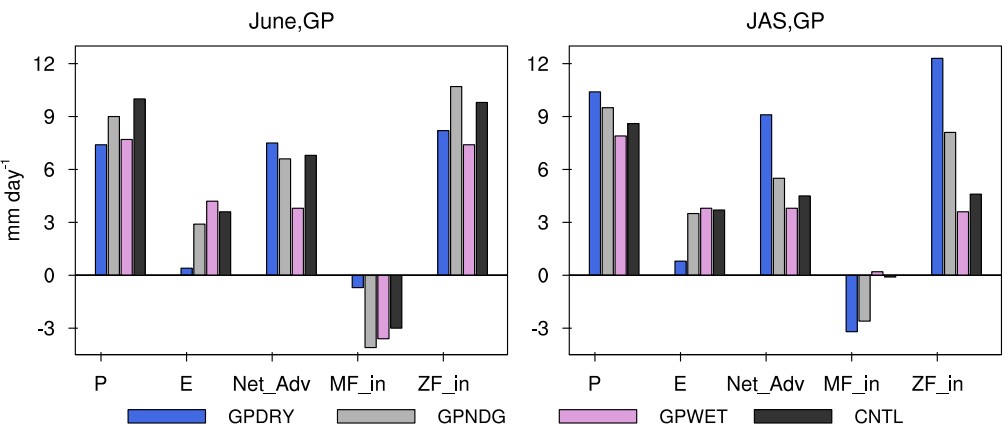

**Figure 7.** Moisture budget components calculated over Gangetic Plain (GP) for GPDRY, GPNDG, GPWET and *CNTL* for, a) June, and b) JAS. P– precipitation, E– evaporation, Net_Adv– net moisture advection into GP, MF_in– net incoming meridional moisture advection, ZF_in– net incoming zonal moisture flux.


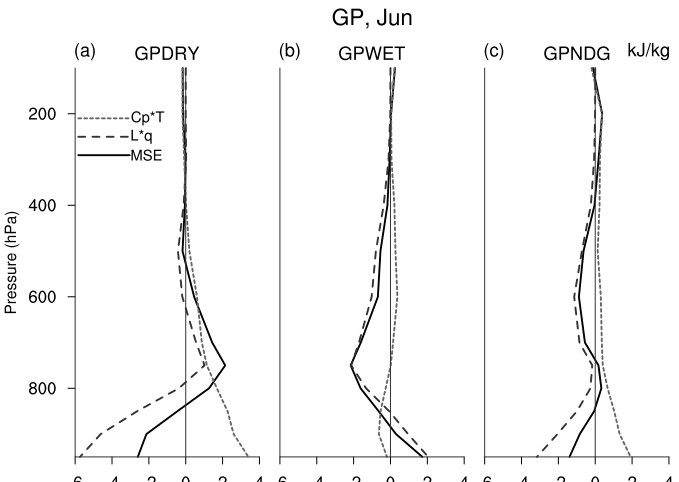

**Figure 8.** Difference in the vertical structure of moist static energy (MSE), moisture ($L\,q$) and temperature ($C_p\,T$) terms (in kJ kg$^{-1}$), area averaged over GP for June, for the three regionally modified runs (over GP) with respect to *CNTL*; a) GPDRY$-$*CNTL*, b) GPWET$-$*CNTL*, c) GPNDG$-$*CNTL*.





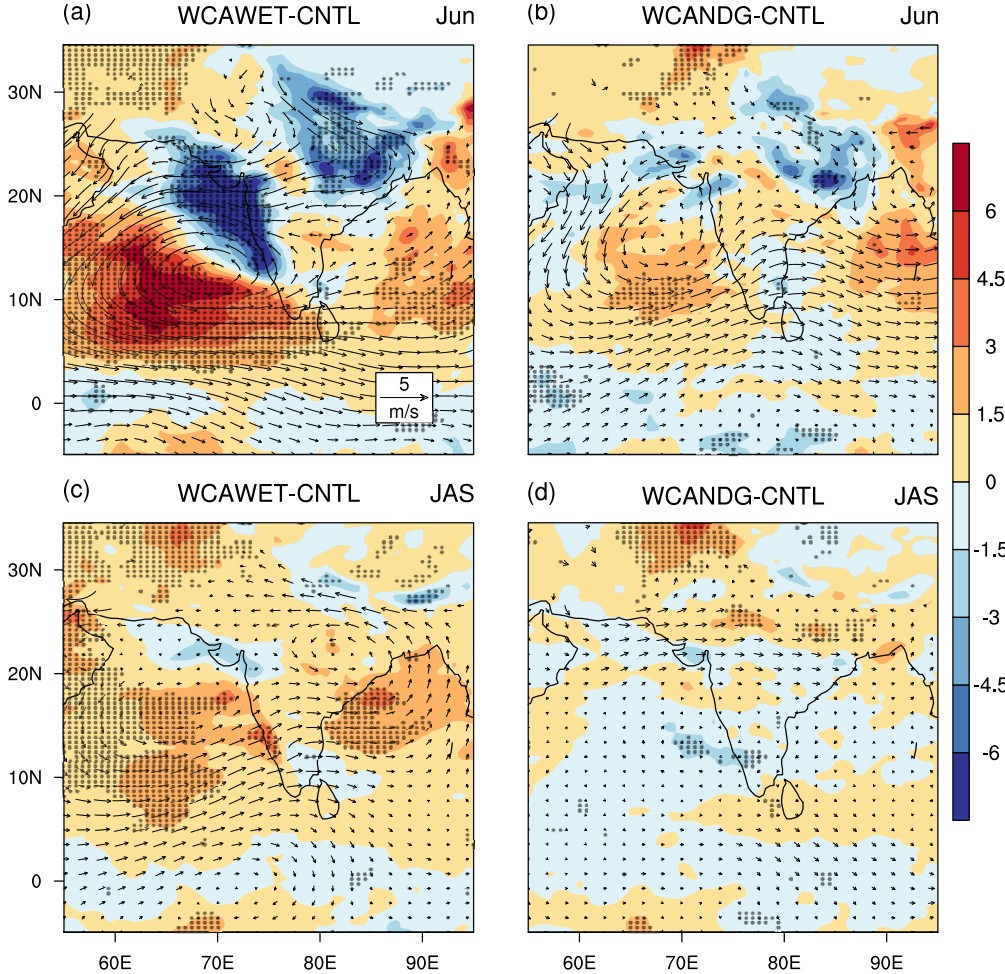

**Figure 9.** Spatial pattern of difference between simulated precipitation (mm day$^{-1}$) and 850 hPa Winds (m s$^{-1}$) from the regionally modified (over WCA) runs and *CNTL* for June and JAS, a) June, WCAWET−*CNTL* b) June, WCANDG−*CNTL* c) JAS, WCAWET−*CNTL* d) JAS, WCANDG−*CNTL*. Precipitation difference significant at 95% level is dotted.





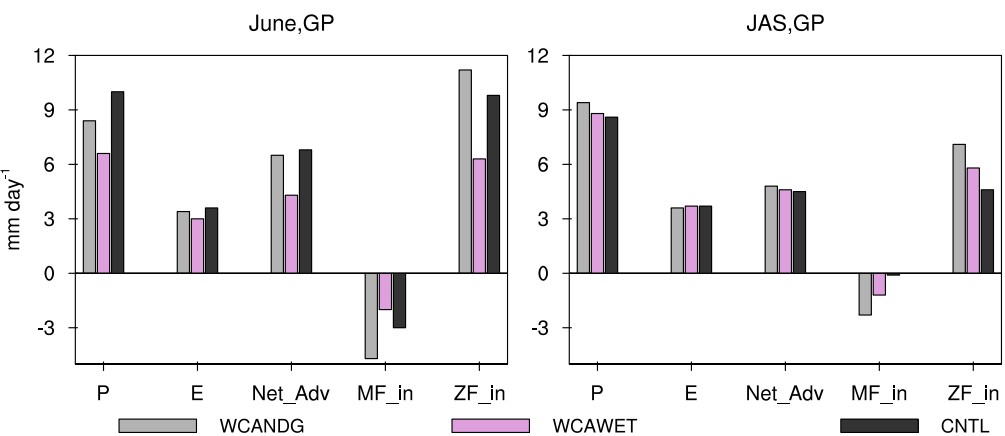

**Figure 10.** Moisture budget components calculated over Gangetic Plain (GP) for WCANDG, WCAWET and *CNTL* for, a) June, and b) JAS.
P– precipitation, E– evaporation, Net_Adv– net moisture advection into GP, MF_in– net incoming meridional moisture advection, ZF_in–
net incoming zonal moisture flux.





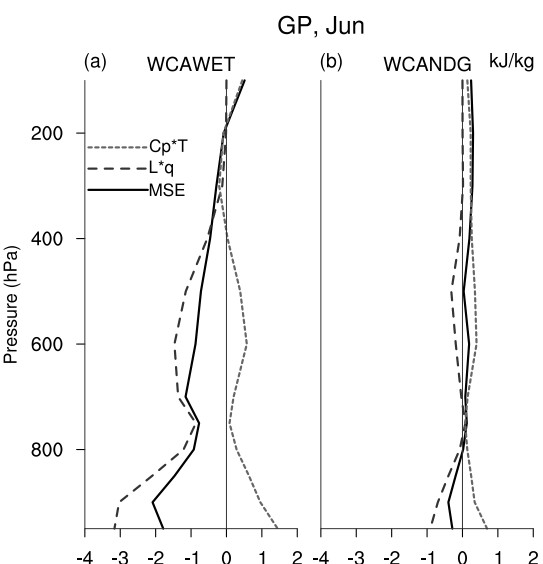

**Figure 11.** Difference in the vertical structure of moist static energy (MSE), moisture ($L\,q$) and temperature ($C_p\,T$) terms (in kJ kg$^{-1}$), area averaged over GP for June, for the two regionally modified runs (over WCA) with respect to *CNTL*; a) WCAWET−*CNTL*, b) WCANDG−*CNTL*.




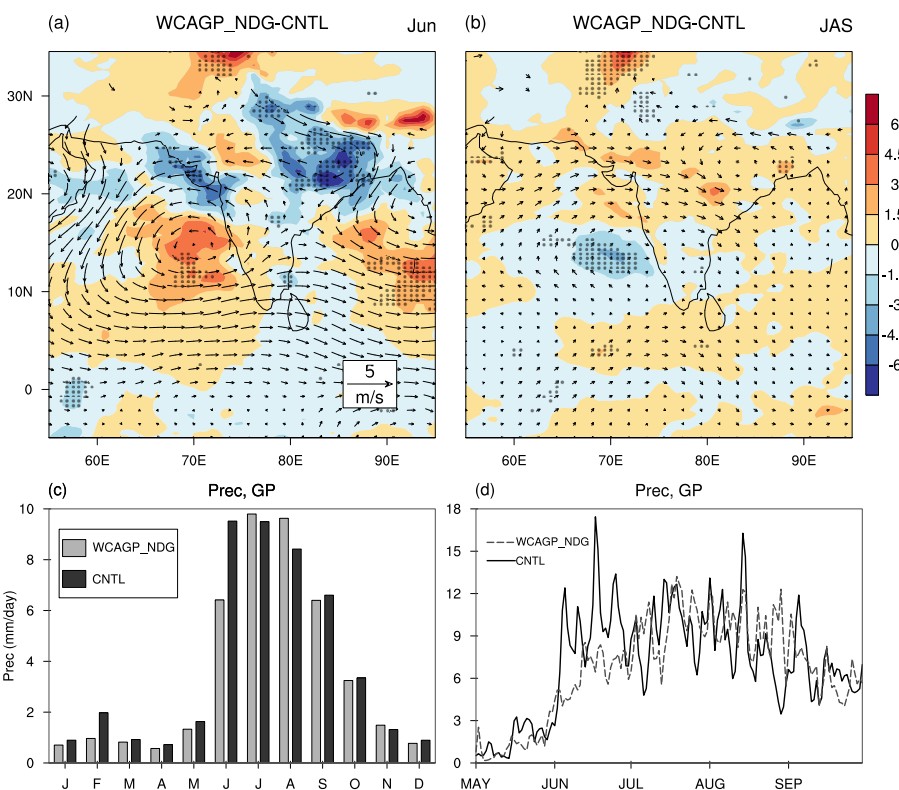

**Figure 12.** Spatial pattern of difference between simulated precipitation (mm day$^{-1}$) and 850 hPa Winds (m s$^{-1}$) from WCAGP_NDG and *CNTL*– a)June, WCAGP_NDG−*CNTL* b) JAS, WCAGP_NDG−*CNTL*. Comparison of climatological rainfall (mm day$^{-1}$) over GP from WCAGP_NDG and *CNTL*, c) monthly means d) seasonal daily mean.





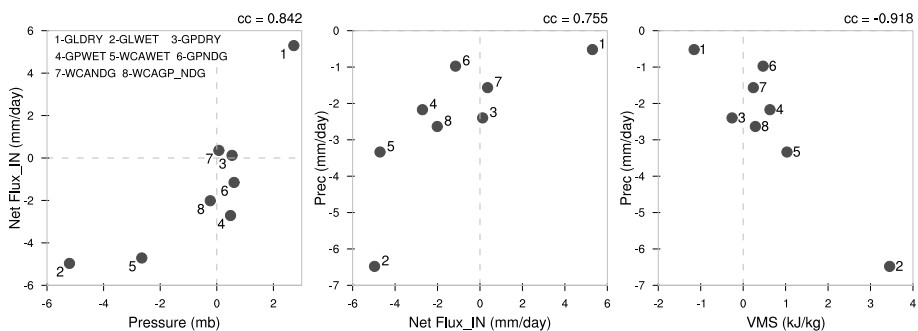

**Figure 13.** Comparison of all the experiments with respect to *CNTL* in June. a) Scatter plot of net incoming moisture flux over GP with respect to pressure difference index (PDI). b) Scatter plot of precipitation over GP with respect to net incoming moisture flux (Net Flux_IN) over GP. c) Scatter plot of precipitation over GP with respect to VMS over GP. PDI is defined as difference of surface pressure (mb) between small region over WCA (55°–72°E and 25°–35°N) and Indian Ocean (55°–72°E and 10°S–0°N).





**Table 1.** List of Experiments, abbreviations and brief descriptions.

| Experiment | Abbreviation | Description |
| --- | --- | --- |
| Control | CNTL | Default configuration |
| Global Dry | GLDRY | SM fixed globally to 1% of saturation |
| Global Wet | GLWET | SM fixed globally to 100% of saturation |
| Gangetic Plain Dry | GPDRY | SM fixed over GP to 1% of saturation |
| Gangetic Plain Wet | GPWET | SM fixed over GP to 100% of saturation |
| Western Central Asia Wet | WCAWET | SM fixed over WCA to 100% of saturation |
| Gangetic Plain Nudged | GPNDG | SM nudged over GP |
| Western Central Asia Nudged | WCANDG | SM nudged over WCA |
| Western Central Asia-Gangetic Plain Nudged | WCAGP_NDG | SM nudged over WCA and GP |



$$MSE = C_p T + Lq + gz \tag{1}$$

$$VMS = MSE_{top} - MSE_{bot} \tag{2}$$

5 $$MSE_{top} = \frac{1}{p_{mid} - p_{top}} \int_{p_{top}}^{p_{mid}} (C_p T + Lq + gz)\, dp \tag{3}$$

$$MSE_{bot} = \frac{1}{p_s - p_{mid}} \int_{p_{mid}}^{p_s} (C_p T + Lq + gz)\, dp \tag{4}$$