# Peer review of "Role of surface hydrology in determining the seasonal cycle of Indian summer monsoon in a general circulation model"

_Hydrology and Earth System Sciences, 2016_

## Referee Comment (RC1) · Anonymous Referee #1 · 26 Dec 2016

This study conducted 10-years simulation of AGCM equipped with CLM to represent precipitation distribution by Indian monsoon system, and try to reveal a role of soil moisture variability in the GCM to modify the summer monsoon onset. According to the comparison of precipitation amount and distribution between CNTL run and TRMM observation, overestimation over the GP was evident on June with positive soil moisture bias in India. According to sensitive studies by changing the soil moisture in different spatial scales, remote dry soil moisture biases over Western Central Asia induced the early monsoon onset in the GP. Combined effects with remote and local configuration of soil moisture could also improve the seasonal progress of Indian monsoon. Authors

concluded that such GCM experiments could contribute for diagnose the function of soil moisture on the monsoon onset and improve the land surface models.

First of all, I would like to mention that reduction of the model bias by changing the boundary condition (such as soil moisture distribution) does not mean that such boundary condition play an important role for actual monsoon variability. This paper confuses those two issues that lead readers for misunderstanding that soil moisture in Western Central Asia play and important role for the monsoon onset in GP. Object of the paper may be limited to improve monsoon simulation in the GCM.

By means of weather and climate system, monsoon precipitation variability in GP area is induced by maritime onset vortex and monsoon lows with continental monsoon trough combined with orographic induced thermal/dynamic flows over Indian sub-continent. If those systems were not represented in the model, precipitation and associated soil moisture distribution are failed. However, the paper did not explain about the representation of precipitation disturbances in the model. For instance, I could not understand that how the anomaly pattern of low-level flow in Fig. 12 could change the precipitation systems. Does the simulation represent diurnal precipitation variability?

I agree that Indian monsoon sector is composed by large land-surface spatial inhomogeneity, such as vegetation, complex terrain and associated soil moisture heterogeneity (P3 upper), that could link to occurrence of individual precipitation systems. Also, diurnal forcing is obviously affecting the precipitation system in the coastal and mountain range areas. To reveal the function of land-atmosphere interaction, non-hydrostatic models with dynamic downscaling simulation, that could explicitly tread the sub-grid scales functions, should be conducted. I understand that computation facility sometimes limits long-term downscaling simulations. However, this paper obviously indicated that GCM is hard to represent a reality associated with sub-grid scale hydrologic heterogeneity, because the CNTL had large biases. I doubt a scientific importance on diagnosing the role of soil moisture impacts by using GCM.

Another problem is the setting of boundary condition in the sensitivity studies. GLDRY and GLWET runs are conducted with 1% and 100 % soil water saturation, respectively. I am not sure how much they could correspond to volumetric water contents by means of in-situ soil structure, but they are definitely unrealistic. The paper set such extreme (idealized) conditions to explain the cause of biases between the CNTL products and actual statuses (observations). By means of idealized simulation, changing of SST in certain areas may cause much impact on GP precipitation, because maritime disturbances have large impacts of northwestern Indian sub-continent climate.

Accordingly, I suggest fundamental revisions before rendering the decision to be published in HESS.

Minor comments P3L7&L16 Is AGCM and CESM the same? Better to unify the term. P3L28 I could not understand the meaning of "nesting LCS4". You calculated the land-surface part within the 0.43*0.63 degree in GCM? Or you nested a certain region in GCM to drive LCS4 ? Section 2.1 How much of the time step for calculation and time resolution of the output? Section 2.2 Better to explain in details about the soil moisture products with reliability. Also, which years of the soil moistures to be used/compared? Fig.1 Why you showed global scale distribution? In the global scale, there are many areas to be explained that CNTL and observations are different except in the Asian monsoon sector. Better to limit the discussion in a monsoon region, such as in Fig. 2. P6L15-25 June is a transitional month from pre-monsoon to monsoon. If the model output could produce daily base, why don't you assess the difference of seasonal progress between the CNTL and observation to identify the key monsoon flow anomaly affected by the differences in soil moisture distribution if any? P8L15-16 A sentence "onset phase and its seasonal cycle" is unclear. What is the "monsoon onset" to be observed in GCM? Fig.6a,b Why the similar precipitation anomaly patterns, such as negative in northeast and positive in the central Indian subcontinent to BOB, even the GPDRY and GPWET setting are opposite soil moisture condition? P8L33 Unclear explanation that "Higher moisture advection" from where to where?

---

## Referee Comment (RC2) · Anonymous Referee #2 · 30 Dec 2016

Role of surface hydrology in determining the seasonal cycle of Indian summer monsoon in a general circulation model by Shubhi Agrawal and Arindam Chakraborty. It is a very interesting work, which highlights effects of soil moisture bias on the simulation Indian summer monsoon rainfall and particularly on the monsoon onset. The manuscript is well written, results are convincing and nicely organized.

This study shows how the excessive dry soil condition (bias) over western Central Asian region can lead to excessive monsoon rainfall during the month of June. On the other hand local soil moisture plays important role during rest of the monsoon

season. This study may have real implications on the understanding of the sources of predictability of the ISMR and hence improving the forecast skill. Overall it will be a good contribution towards our understanding of land-atmosphere interactions over the south Asian monsoon region. These are comments (minor), which needs to be addressed before it is accepted for publication.

1) The result of this modeling study is very much consistent with Rai et al. (2015), which is based on only observations. Pre-onset (April-May) rainfall and 2m air temperature, which can be also used as proxy for soil moisture/land-surface conditions, shows strong inverse link with the first phase (June-July) of monsoon rainfall (see Figure 1,2 in Rai et al 2015). Similarly there are previous study by Parthasarathy et al., 1992; Singh et al., 1995 . The point is here that this modeling study should be build on such note and finally this study shows that model is able to capture this observed teleconnection faithfully.

2) Many previous observational study have pointed out that pre-onset land surface conditions, particularly over the heat-low regions (Iran, Afghanistan, Arabian region) have significant impact on the performance of ISMR. This could be one of the non-ENSO sources of predictability in a forecast model. Unfortunately we do not have deeper layer soil moisture observation, which can be feed into land data assimilation system.

3) In the summary and conclusions part "It follows from our work that the surface soil moisture anomalies bear serious consequences ........ " It is not anomaly but a systematic bias.

4) Soil moisture shown in Figure 5 is from top model layer ? In nudged experiment, only top soil layer is nudged ? What about the deeper layers, what kind of effects it can have on the results ?

5) In page 2, " The contribution of land-atmosphere interaction......." is a very big sentence, hard to read. Please split it into smaller sentences.

Parthasarathy, B., R. Kumar, and A. A. Munot (1992), Surface pressure and summer monsoon rainfall over India, Adv. Atmos. Sci., 9, 359–366.

Singh, D., C. V. V. Bhadram, and G. S. Mandal (1995), New regression model for Indian summer monsoon rainfall, Meteorol. Atmos. Phys., 55, 77–86.

---

## Referee Comment (RC3) · Anonymous Referee #3 · 2 Jan 2017

General comments: This study starts with showing the precipitation bias between model results and observation datasets during the Indian summer monsoon period, combining with analyzing the vertical moisture stability. Furtherly, the sensitive experiments are conducted to indicate that the effect of remote soil moisture over the WCA contribute more to the precipitation bias in June than that of local ones over the GP. However, about description of the moisture circulation in the remote influence is very confused. First of all, due to the location of GP, how could the intensifying low-level westerly jet (depicted in Fig.9b) influenced by the negative soil moisture over the WCA bring more moisture to GP in June? The southwest wind cannot reach to GP and the

wind interacting area locates in the southern part to GP. In addition, the explanation of the low-level jet in the sensitive experiment of WCAGP_NDG should be specific, so does ITCZ. Overall, the soil moisture, especially explaining from its induced moisture circulation, cannot be seen that it can improve the prediction of the onset of the Indian summer monsoon in this study. But the results are promising to advance the GCM simulating the Indian summer monsoon precipitation.

Specific comments: 1) Page 6 line 20, 'that is an onset is declared if the area averaged rainfall is more that 4 mm day$-1$ for at least five consecutive days after first of June'. This sentence need to be rewrittern. 2) The usage of which and that is not proper, eg. Page 8 line 28 'under extreme conditions, that is very dry surface and completely saturated surface' where' that' should be changed to 'which'. 3) Page 9 line 9, 'though this small. . ., but. . .' where but should be removed. 4) Page 9 line 14, '. . .weakens the circulation. . .'where 'the circulation' is not very clear. 5) Page 13 line 6, '. . .modulate the onset phase and seasonal cycle of Indian summer monsoon' where 'the onset phase' should be specific. 6) About the equations, please give the details of the physical meaning of every alphabet.

---

## Author Comment (AC1) · 8 Feb 2017

MS No.: hess-2016-591

**Reply to Referee 1 on the manuscript "Role of surface hydrology in determining the seasonal cycle of Indian summer monsoon in a general circulation model"**

Shubhi Agrawal[1] and Arindam Chakraborty[1,2]

[1]Centre for Atmospheric and Oceanic Sciences, Indian Institute of Science, Bengaluru, India
[2]Divecha Centre for Climate Change, Indian Institute of Science, Bengaluru, India

First of all, authors thank the reviewer for fundamental comments on this study.

*Comment: This study conducted 10-years simulation of AGCM equipped with CLM to represent precipitation distribution*

5  *by Indian monsoon system, and try to reveal a role of soil moisture variability in the GCM to modify the summer monsoon onset. According to the comparison of precipitation amount and distribution between CNTL run and TRMM observation, overestimation over the GP was evident on June with positive soil moisture bias in India. According to sensitive studies by changing the soil moisture in different spatial scales, remote dry soil moisture biases over Western Central Asia induced the early monsoon onset in the GP. Combined effects with remote and local configuration of soil moisture could also improve*

10  *the seasonal progress of Indian monsoon. Authors concluded that such GCM experiments could contribute for diagnose the function of soil moisture on the monsoon onset and improve the land surface models.*

*First of all, I would like to mention that reduction of the model bias by changing the boundary condition (such as soil moisture distribution) does not mean that such boundary condition play an important role for actual monsoon variability. This paper confuses those two issues that lead readers for misunderstanding that soil moisture in Western Central Asia play and*

15  *important role for the monsoon onset in GP. Object of the paper may be limited to improve monsoon simulation in the GCM.*

Reply: We strongly disagree with the reviewer. Firstly, the purpose of this study is NOT to illustrate that a prescribed field of boundary condition (such as soil moisture) in the model improves the simulated (interannual) variability of monsoon. Our manuscript neither does discuss interannual variation of model simulated results in its default format (CNTL) nor indicates any change in its interannual variations resulting from the perturbed simulations. Instead, we emphasise that both local and

20  remote soil moisture biases simulated by the model contribute to distorted seasonal cycle of monsoon over GP. For that, we did a 10-year long simulation without year-to-year variation of SST, to eliminate the interannual variation arising from it.

We agree that our study shows the importance of soil moisture bias over Western Central Asia (WCA) on the early monsoon rainfall (possibly linked to onset of monsoon) over GP in this model. We would like to refer to a few recent studies showing the importance of climate over WCA on monsoon over India. Samson et al. (2016) shows the importance of prescribed albedo over

25  WCA on annual cycle. However, in a model surface albedo is a function of soil moisture. And thus, our study both confirms the finding of Samson et al. (2016) and takes it a step ahead in the context of error in another climate model. Rai et al. (2015),

using observations, have shown the importance of climate over WCA during April-May on the early monsoon rainfall over India. We have shown that error in land-surface model can introduce large error in the simulation of rainfall annual cycle. And thus our study hints at the need to improve surface processes in the CLM, especially over GP and WCA. And while combining these above findings, we feel that, once this bias is improved, it will help improve the interannual variation of monsoon in the model. However, this could be a scope of future study.

*Comment: By means of weather and climate system, monsoon precipitation variability in GP area is induced by maritime onset vortex and monsoon lows with continental monsoon trough combined with orographic induced thermal/dynamic flows over Indian sub-continent. If those systems were not represented in the model, precipitation and associated soil moisture distribution are failed. However, the paper did not explain about the representation of precipitation disturbances in the model. For instance, I could not understand that how the anomaly pattern of low-level flow in Fig. 12 could change the precipitation systems. Does the simulation represent diurnal precipitation variability?*

We understand the concern of the reviewer, that a poor representation of the oceanic systems such as lows would impact the rainfall, and subsequently soil moisture, over GP. Such systems are primarily concentrated during the monsoon season. We would like to refer to Figures 2c and 5 of our manuscript, where we show the annual cycle of rainfall over GP as well as soil moisture over GP and WCA. A careful comparison reveals that CNTL has positive (negative) soil moisture bias over GP (WCA) throughout the year. This is in spite of comparable amount of rainfall in CNTL to that observed in March–May over GP (Fig 2c). Thus, this bias in soil moisture during pre-monsoon season cannot be due to error in representing monsoon systems by the model. Similarly, a consistent dry bias over semi-arid region like WCA calls for further investigation of its cause in the model, including its land-surface scheme.

In accordance of this comment of the reviewer, however, we have looked at the skill of CNTL in representing intraseasonal variability of rainfall which are tightly associated with monsoon systems. Such analysis can best be done in frequency domain similar to that carried out by Karmakar et al. (2015). Two dominant characteristic time-scales of variability of Indian monsoon are the low frequency oscillations (on 20-60 days time-scale) and the high frequency oscillations (on 10-20 days time-scale). The model is able to capture the variability in both the modes reasonably. Following figure summarizes the percentage of the total daily variability explained by the low frequency and high frequency oscillations in TRMM estimation and CNTL simulations over GP. We plan to include a discussion on this in the revised manuscript.

Dai and Trenberth (2004) showed that CAM2 was able to capture the diurnal cycle of surface air temperature well, especially over land. But this model suffered from systematic bias in cloud fraction simulation. This deficiency was mainly linked to convective parameterization scheme. Yuan et al.(2013) showed that CAM5 was able to capture the diurnal cycle of precipitation over East Asia, with varying accuracy spatially. CAM5 simulated the diurnal variation of stratiform clouds better as compared to convective clouds. Moreover, the model was also able to capture the diurnal phase variations of low level winds. We have not looked into diurnal time-scales in this work, which focuses on biases in the monthly and seasonal means.

*Comment: I agree that Indian monsoon sector is composed by large land-surface spatial inhomogeneity, such as vegetation, complex terrain and associated soil moisture heterogeneity (P3 upper), that could link to occurrence of individual precipitation systems. Also, diurnal forcing is obviously affecting the precipitation system in the coastal and mountain range areas. To*

[Figure]

**Figure 1.** Contribution of the low-frequency intraseasonal oscillations (LF-ISO) and the high-frequency intraseasonal oscillations (HF-ISO) to the total daily variance of rainfall in June–September over Gangetic Plain. The error bars show one interannual standard deviation.

*reveal the function of land-atmosphere interaction, non- hydrostatic models with dynamic downscaling simulation, that could explicitly tread the sub-grid scales functions, should be conducted. I understand that computation facility sometimes limits long-term downscaling simulations. However, this paper obviously indicated that GCM is hard to represent a reality associated with sub-grid scale hydrologic heterogeneity, because the CNTL had large biases. I doubt a scientific importance on diagnosing the role of soil moisture impacts by using GCM.*

Reply: Please note that we have used a global climate model at a resolution that is reasonably high (about 0.5 degrees in longitude/latitude) to represent large scale heterogeneity in land surface and orography. Moreover, the land component of our model, CLM, has robust schemes to deal with sub-grid level heterogeneity. We have discussed this in Page 3, Line 28 (Section 2.1 of the manuscript) along with relevant references.

A non-hydrostatic model demands a resolution that is often impossible to run in climate mode (such as that in this study) even with fastest computers. Thus, use of such model would require a compromise to use it only over a specific region. We would like to point here out that such regional models, since forced at its lateral boundaries, are unable to consider atmospheric feedback arising out from remote locations. One example could be global scale change in upper tropospheric circulation due to regional forcing. These feedbacks become especially important in climate time scales. Therefore, we do not prefer to use a regional model to address the problem of this study.

*Comment: Another problem is the setting of boundary condition in the sensitivity studies. GLDRY and GLWET runs are conducted with 1% and 100% soil water saturation, respectively. I am not sure how much they could correspond to volumetric water contents by means of in-situ soil structure, but they are definitely unrealistic. The paper set such extreme (idealized) conditions to explain the cause of biases between the CNTL products and actual statuses (observations).*

Reply: Firstly, we would like to clarify the point relating soil water and volumetric water content. A 100% saturation (SAT runs) means filling the soil with its field capacity. This corresponds to $0.48\ mm^3/mm^3$ over GP region. Similarly, 1% saturation (DRY runs) corresponds to about $0.005\ mm^3/mm^3$ of water in volumetric units.

Secondly, regarding the unrealistic nature of these two simulations (GLDRY and GLWET), we agree that these are extreme cases. However, we have not used these two simulations to explain the cause of bias in CNTL. We have only used these two runs at the end of paper, in Figure 13, to show the sensitivity of simulated precipitation in model to soil moisture as boundary condition. These two runs set the extremes of sensitivity of model to soil moisture conditions. And it can be noted from Figure 13 that all other experiments fall between these two experiments.

*Comment: By means of idealized simulation, changing of SST in certain areas may cause much impact on GP precipitation, because maritime disturbances have large impacts of northwestern Indian sub-continent climate.*

Reply: We agree that changing SSTs in certain areas may have impact on GP precipitation. But the goal of this study is to analyse the effect of land-atmosphere coupling over GP. And this is best understood when SST variations do no influence the land-atmosphere interaction.

**Minor comments**

*1. P3L7&L16 Is AGCM and CESM the same? Better to unify the term.*

Reply: We have used the atmospheric component of CESM model in our study, which is referred as CAM in manuscript. We will replace the term "AGCM" appropriately to avoid confusion.

*2. P3L28 I could not understand the meaning of "nesting LCS4". You calculated the land- surface part within the 0.43\*0.63 degree in GCM? Or you nested a certain region in GCM to drive LCS4 ?*

Reply: We have not used the term "nesting LCS4" anywhere in the manuscript. But clarification about "nested subgrid level hierarchy" in P3L28 is as follows: we have run the CLM model at a grid resolution of $0.43 \times 0.63$ degree which is same as that for its atmospheric counterpart. The phrase "nested subgrid level hierarchy" in description of CLM model refers the way in which sub-grid level heterogeneity is taken care of in the model.

*3. Section 2.1 How much of the time step for calculation and time resolution of the output?*

Reply: Section 2.1 is related to model description. We will add a line about time-step and output frequency in section 2.3-Experiment setup details in the beginning of section. This information was added under "Regionally Nudged Runs" - P5L6 and P5L13.

*4. Section 2.2 Better to explain in details about the soil moisture products with reliability. Also, which years of the soil moistures to be used/compared?*

Reply: ESACCI Soil moisture data for the period 2000-2010 has been used (P4L16). The manuscript includes references (Wagner et al., 2012; Liu et al., 2012) for details on ESACCI data product. It also includes a reference for ESACCI data validation carried out by Dorigo et al. (2015), using ground-based observations. We will add more details about the ESACCI data and its reliability in the manuscript.

*5. Fig.1 Why you showed global scale distribution? In the global scale, there are many areas to be explained that CNTL and observations are different except in the Asian monsoon sector. Better to limit the discussion in a monsoon region, such as in Fig. 2.*

Reply: We have shown Asia-Pacific monsoon region (Wang and LinHo, 2002) in Figure 1, (that is 15 S– 45 N, 40– 290 E). We wanted to highlight that model captures the spatial distribution of seasonal mean rainfall (Jun-Sep) reasonably well over Asia-Pacific monsoon system, with a spatial correlation of 0.76. This figure shows the large scale picture. We also show the monthly mean rainfall over Indian region to highlight the bias in precipitation in model over Indian region. Thereon we move towards the objective of this paper.

*6. P6L15-25 June is a transitional month from pre-monsoon to monsoon. If the model output could produce daily base, why don't you assess the difference of seasonal progress between the CNTL and observation to identify the key monsoon flow anomaly affected by the differences in soil moisture distribution if any? P8L15-16 A sentence "onset phase and its seasonal cycle" is unclear. What is the "monsoon onset" to be observed in GCM?*

Reply: We thank the reviewer for his/her views on this seasonal progress. We would like to point out that CNTL simulated soil moisture over GP was higher than that observed throughout the year (Fig 5c). However, bias in CNTL simulated precipitation over GP was highest in June (Fig 2), that decreases in the subsequent months. In this study we have separated out June and July-September to show this seasonal progress of monsoon over India, especially over GP.

By "onset phase of monsoon over India" we mean early part of June month, when monsoon sets in over the southern tip of India and start progressing north-westward to cover the country in another about a month.

By "seasonal cycle" we mean seasonal cycle of Indian monsoon, that is related to seasonal northward migration of the intertropical covergence zone (ITCZ) up to the foothills of India.

Monsoon onset over GP has been defined in P6L19 - "We calculate onset date of monsoon over GP for CNTL and TRMM following the criterion used by Chakraborty et al. (2006), that is an onset is declared if the area averaged rainfall is more that 4 mm day$^{-1}$ for at least five consecutive days after first of June". We will rewrite this sentence for more clarity.

*7. Fig.6a,b Why the similar precipitation anomaly patterns, such as negative in northeast and positive in the central Indian subcontinent to BOB, even the GPDRY and GPWET setting are opposite soil moisture condition?*

Reply: We have explained the cause of negative precipitation anomaly over GP (extending towards northeast) through moisture budget analysis in Fig. 7a (and explained in P9L3-7). In June (Fig. 7a), for GPDRY,net moisture advection slightly increased and evaporation reduced substantially, overall reducing precipitation compared to CNTL. For GPWET, net moisture advection reduced substantially and evaporation increased slightly , again reducing the precipitation compared to CNTL. Regarding positive anomalies over BOB, these are related to changes in wind circulation mainly (850 hPa wind differences are shown in Fig.6). Fig.6a,b show intensification of winds south of GP, extending over BOB. Southward shift in rainbands can cause this increase in precipitation over BOB, as compared to CNTL.

*8. P8L33 Unclear explanation that "Higher moisture advection" from where to where?*

Reply: Here we explain the low level circulation changes in GPDRY experiment as compared to CNTL. By "higher moisture advection" we mean in GPDRY, more moisture is advected into GP from surrounding areas, as compared to CNTL. We will modify statement for clarity.

**References:**

5    Karmakar, N., A. Chakraborty, and R. S. Nanjundiah, 2015: Decreasing intensity of monsoon low-frequency intraseasonal variability over India. Environ. Res. Lett., 10, 054018, doi:10.1088/1748-9326/10/5/054018.

Samson, G., Masson, S., Durand, F., Terray, P., Berthet, S., and Jullien, S.: Roles of land surface albedo and horizontal resolution on the Indian summer monsoon biases in a coupled ocean–atmosphere tropical-channel model, Climate Dynamics, pp. 1–24, doi:10.1007/s00382-016- 3161-0, http://dx.doi.org/10.1007/s00382-016-3161-0, 2016.

10   Yuan, W., R. Yu, M. Zhang, W. Lin, J. Li, and Y. Fu, 2013: Diurnal cycle of summer precipitation over subtropical East Asia in CAM5. J. Climate, 26, 3159–3172, doi:10.1175/JCLI-D-12-00119.1.

Dai, A., and K. E. Trenberth, 2004: The diurnal cycle and its depiction in the Community Climate System Model. J. Climate, 17, 930–951, doi:10.1175/1520-0442(2004)017<0930:TDCAID>2.0.CO;2.

---

## Author Comment (AC2) · 8 Feb 2017

MS No.: hess-2016-591

**Reply to Referee 2 on the manuscript "Role of surface hydrology in determining the seasonal cycle of Indian summer monsoon in a general circulation model"**

Shubhi Agrawal[1] and Arindam Chakraborty[1,2]

[1]Centre for Atmospheric and Oceanic Sciences, Indian Institute of Science, Bengaluru, India
[2]Divecha Centre for Climate Change, Indian Institute of Science, Bengaluru, India

Thank you for your valuable feedback and suggestions and your time!

*Comment: Role of surface hydrology in determining the seasonal cycle of Indian summer monsoon in a general circulation*

5   *model by Shubhi Agrawal and Arindam Chakraborty. It is a very interesting work, which highlights effects of soil moisture bias on the simulation Indian summer monsoon rainfall and particularly on the monsoon onset. The manuscript is well written, results are convincing and nicely organized. This study shows how the excessive dry soil condition (bias) over western Central Asian region can lead to excessive monsoon rainfall during the month of June. On the other hand local soil moisture plays important role during rest of the monsoon season. This study may have real implications on the understanding of the*

10  *sources of predictability of the ISMR and hence improving the forecast skill. Overall it will be a good contribution towards our understanding of land-atmosphere interactions over the south Asian monsoon region.*

Reply: Thank you for sharing your thoughts on the implication of this work!

*Comment: These are comments (minor), which needs to be addressed before it is accepted for publication. 1) The result of this modeling study is very much consistent with Rai et al. (2015), which is based on only observations. Pre-onset (April-May)*

15  *rainfall and 2m air temperature, which can be also used as proxy for soil moisture/land-surface conditions, shows strong inverse link with the first phase (June-July) of monsoon rainfall (see Figure 1,2 in Rai et al 2015).*

Reply: We have already discussed this relevant paper of Rai et al. (2015) in the Introduction section (P2L5). In the revised manuscript, we plan to discuss further results from this paper, similar to our model simulation results.

*Similarly there are previous study by Parthasarathy et al., 1992; Singh et al., 1995 . The point is here that this modeling study*

20  *should be build on such note and finally this study shows that model is able to capture this observed teleconnection faithfully.*

Reply: Thank you for these references. We will include these references in relevant result sections.

*2) Many previous observational study have pointed out that pre-onset land surface conditions, particularly over the heat-low regions (Iran, Afghanistan, Arabian region) have significant impact on the performance of ISMR. This could be one of the non-ENSO sources of predictability in a forecast model.*

Reply: This goes in line with the future applicability of the present study. Our results in this paper are based on model simulations with climatological SSTs and still we see a significant change in seasonal cycle over Gangetic Plains, highlighting the importance of land-atmosphere coupling over this region.

*Unfortunately we do not have deeper layer soil moisture observation, which can be feed into land data assimilation system.*

Reply: We agree with the reviewer on this. More observational data from deeper soil layers, like soil moisture, soil temperature, infiltration, etc, will be very useful in evaluating and improving land model.

*3) In the summary and conclusions part "It follows from our work that the surface soil moisture anomalies bear serious consequences ........ " It is not anomaly but a system- atic bias.*

Reply: Thank you for your suggestion. We will modify this as per suggestion.

*4) Soil moisture shown in Figure 5 is from top model layer? In nudged experiment, only top soil layer is nudged? What about the deeper layers, what kind of effects it can have on the results ?*

Reply: Yes, soil moisture shown in Figure 5 is from top layer of model. Depths of top two layers in the model are 0.007100635m, 0.027925m. In the nudged runs these top two layers are nudged, as the satellite based observation based volumetric soil moisture data (ESACCI) represents a depth between 0.005-0.02m.

Vertical profile of soil column over GP (area averaged soil moisture) is plotted against day of the year in below Figure. This figure shows the effect of nudging over GP on the deeper soil layer. Soil moisture values for deeper layer are marginally different in the CNTL and WCAGP_NDG experiments.

[Figure]

**Figure 1.** Vertical profile of volumetric Soil moisture area averaged over Gangetic plain (76–88 E, 22–28 N) plotted against calendar days for CNTL (top panel) and WCAGP_NDG (bottom panel).

*5) In page 2, " The contribution of land-atmosphere interaction......." is a very big sentence, hard to read. Please split it into smaller sentences.*

Reply: We will modify the sentence as per suggestion to make it more clear.

---

## Author Comment (AC3) · 8 Feb 2017

MS No.: hess-2016-591

**Reply to Referee 3 on the manuscript "Role of surface hydrology in determining the seasonal cycle of Indian summer monsoon in a general circulation model"**

Shubhi Agrawal[1] and Arindam Chakraborty[1,2]

[1]Centre for Atmospheric and Oceanic Sciences, Indian Institute of Science, Bengaluru, India
[2]Divecha Centre for Climate Change, Indian Institute of Science, Bengaluru, India

Thank you for your valuable feedback and suggestions and your time!

*General comments: This study starts with showing the precipitation bias between model results and observation datasets*

5 *during the Indian summer monsoon period, combining with analyzing the vertical moisture stability. Furtherly, the sensitive experiments are conducted to indicate that the effect of remote soil moisture over the WCA contribute more to the precipitation bias in June than that of local ones over the GP. However, about description of the moisture circulation in the remote influence is very confused.*

Reply: We will modify the remote forcing section and make it more elaborate and clear.

10 *Comment: First of all, due to the location of GP, how could the intensifying low-level westerly jet (depicted in Fig.9b) influenced by the negative soil moisture over the WCA bring more moisture to GP in June? The southwest wind cannot reach to GP and the wind interacting area locates in the southern part to GP.*

Reply: In our paper, the Gangetic Plain (22–28 N and 76–88 E) is very close to monsoon core zone. In Figure 9a, where we have shown the difference of precipitation and 850 hPa winds between WCAWET and CNTL (WCAWET-CNTL), the

15 anti-cyclonic nature of 850 hPa winds over GP in WCAWET with respect to CNTL is noticeable. WCAWET experiment has wetter soil moisture condition over WCA compared to CNTL. The point we want to make here is that the moisture influx into GP depends on the cyclonic turning of low level winds, which also depends on the strength of the south-westerly Jet.

Additionally, the figure below shows a weakening of the 850 hPa westerlies over 65-75 E in June in WCAWET run as compared to control.

[Figure]

**Figure 1.** Zonal wind at 850 hPa averaged between 65–75 E plotted versus latitude for the month of June for WCAWET and CNTL.

*Comment: In addition, the explanation of the low-level jet in the sensitive experiment of WCAGP_NDG should be specific, so does ITCZ.*

Reply: We will modify appropriately. Sorry for the confusion.

*Comment: Overall, the soil moisture, especially explaining from its induced moisture circulation, cannot be seen that it can*
5 *improve the prediction of the onset of the Indian summer monsoon in this study. But the results are promising to advance the GCM simulating the Indian summer monsoon precipitation.*

Reply: We tend to disagree here with the reviewer. In this work we do not want to claim that soil moisture nudging improved the prediction of onset. We only want to bring to the attention of readers the improvement in seasonal cycle of precipitation over GP with soil moisture bias correction, especially in early June, that is onset phase of monsoon. But we neither deny the
10 implication that this link between WCA soil moisture anomalies and monsoon onset over India could be of use in improving prediction of onset, but it is surely outside the scope of this study.

Our results show that soil moisture bias in model is responsible for distorted seasonal cycle and sharp rise in precipitation after onset over GP in beginning of June. Towards the end of paper, we demonstrate through WCAGP_NDG experiment, that after soil moisture bias correction over WCA and GP, the seasonal cycle over GP improved, and also the sharp rise in
15 precipitation is not there, and the precipitation gradually increases in June. We do not want to assert through our experiments that onset date is changed through soil moisture bias correction. We only want to highlight the sharp increase in precipitation

in CNTL after onset. We will modify the sentences in manuscript, which could cause such confusion to readers. We are sorry for this confusion in the first draft of manuscript. We will make it more clear.

*Specific comments: 1) Page 6 line 20, " that is an onset is declared if the area averaged rainfall is more that 4 mm day$^{-1}$ for at least five consecutive days after first of June ". This sentence need to be rewrittern.*

Reply: We will rewrite this sentence.

*2) The usage of which and that is not proper, eg. Page 8 line 28 "under extreme conditions, that is very dry surface and completely saturated surface" where" that" should be changed to "which".*

Reply: We will modify appropriately.

*3) Page 9 line 9, 'though this small. . ., but. . .' where but should be removed.*

Reply: We will correct it.

*4) Page 9 line 14, '. . .weakens the circulation. . .'where 'the circulation' is not very clear.*

Reply: By 'the circulation' , we basically mean moisture influx over GP in CNTL. We have pointed out the net moisture advection and net zonal moisture advection terms in previous line. But we will modify this sentence for clarity.

*5) Page 13 line 6, '. . .modulate the onset phase and seasonal cycle of Indian summer monsoon' where 'the onset phase' should be specific.*

Reply: By onset phase we mean early part of June. Major difference in seasonal cycle of precipitation over GP between CNTL and WCAGP_NDG is noticeable (Figure 12d). We will further modify in manuscript for clarity.

*6) About the equations, please give the details of the physical meaning of every alphabet.*

Reply: We will add details. Thank you for your comments.